# From Synaptic Plasticity to Neurodegeneration: BDNF as a Transformative Target in Medicine

**DOI:** 10.3390/ijms26094271

**Published:** 2025-04-30

**Authors:** Corneliu Toader, Matei Serban, Octavian Munteanu, Razvan-Adrian Covache-Busuioc, Mihaly Enyedi, Alexandru Vlad Ciurea, Calin Petru Tataru

**Affiliations:** 1Department of Neurosurgery, “Carol Davila” University of Medicine and Pharmacy, 020021 Bucharest, Romania; corneliu.toader@umfcd.ro (C.T.); matei.serban2021@stud.umfcd.ro (M.S.); razvan-adrian.covache-busuioc0720@stud.umfcd.ro (R.-A.C.-B.); prof.avciurea@gmail.com (A.V.C.); 2Department of Vascular Neurosurgery, National Institute of Neurology and Neurovascular Diseases, 077160 Bucharest, Romania; 3Puls Med Association, 051885 Bucharest, Romania; 4Department of Anatomy, “Carol Davila” University of Medicine and Pharmacy, 050474 Bucharest, Romania; mihaly.enyedi@umfcd.ro; 5Neurosurgery Department, Sanador Clinical Hospital, 010991 Bucharest, Romania; 6Medical Section, Romanian Academy, 010071 Bucharest, Romania; 7Department of Opthamology, “Carol Davila” University of Medicine and Pharmacy, 020021 Bucharest, Romania; calin.tataru@umfcd.ro

**Keywords:** BDNF, synaptic plasticity, precision medicine, Alzheimer’s disease, CRISPR, neurotrophic signaling, personalized therapies, TrkB agonists

## Abstract

The brain-derived neurotrophic factor (BDNF) has become one of the cornerstones of neuropathology, influencing synaptic plasticity, cognitive resilience, and neuronal survival. Apart from its molecular biology, BDNF is a powerful target for transformative benefit in precision medicine, leading to innovative therapeutic approaches for neurodegenerative and psychiatric diseases like Alzheimer’s disease (AD), Parkinson’s disease (PD), major depressive disorder (MDD), and post-traumatic stress disorder (PTSD). Nevertheless, clinical applicability is obstructed by hurdles in delivery, patient-specific diversity, and pleiotropic signaling. Here, we summarize findings in BDNF research, including its regulatory pathways and diagnostic/prognostic biomarkers and integrative therapeutic approaches. We describe innovative delivery systems, such as lipid nanoparticle-based mRNA therapies and CRISPR-dCas9-based epigenetic editing that bypass obstacles such as BBB (blood–brain barrier) and enzymatic degradation. The recent implementation of multiplex panels combining BDNF biodynamic indicators with tau and amyloid-β signaling markers showcases novel levels of specificity for both early detection and potential therapeutic monitoring. Humanized preclinical models like iPSC-derived neurons and organoids point to the key role of BDNF in neurodeveloping and neurodegenerative processes, paralleling advances in bridging preclinical observation and clinical environments. Moreover, novel therapeutic tools delivering TrkB activators or the implementation of AI-based dynamic care platforms enable tailored and scalable treatments. This review also aims to extend a framework used in the understanding of BDNF’s relevance to traditional neurodegenerative models by situating more recent work detailing BDNF’s actions in ischemic tissues and the gut–brain axis in the context of systemic health. Finally, we outline a roadmap for the incorporation of BDNF-centered therapies into worldwide healthcare, highlighting ethical issues, equity, and interdisciplinary decomposition. The therapeutic potential of BDNF heralds a new era in neuroscience and medicine, revolutionizing brain health and paving the way for the advancement of precision medicine.

## 1. Introduction

### 1.1. Unveiling BDNF: A Pillar of Neurobiology

Brain-derived neurotrophic factor (BDNF) is a core area of today’s neurobiology. It modulates related cellular processes in neurons, including survival, differentiation, and synaptic plasticity. BDNF is a polyfunctional neurotrophin, like nerve growth factor (NGF) or neurotrophin-3 (NT-3), and has firmly established itself, after its initial discovery by Yves-Alain Barde et al. We know that activity dependently as well as using several regulators that are derived from neuronal activity and environmental adaptations BDNF is released to the local area of the neurons that activated, binds, activates and delivers its actions via its receptor. This atypical action provides a unique basis for BDNF to serve as a molecular learning modulator as well as a chief organizer of learning, memory, and adaptive behavior [1,2].

BNDF is expressed with great abundance in both the central and peripheral nervous systems, congruent with its functional wealth. Overall, BDNF secreted during the critical period of development promotes new immature neuron survival and directs the differentiation of specialized cell types and excitatory and inhibitory subtypes, being the fundamental unit of definition of increasingly complex neuronal circuitry [3]. In adulthood, BDNF gives a stable synaptic identity, promotes synaptic plasticity, and encourages long-term potentiation (LTP) and long-term depression (LTD)—two forms of training, memory consolidation, and behavioral flexibility in the brain. Thus, with both neurogenesis and synaptic remodeling, BDNF has established itself as a convening point between basic neurobiology and translational research in neuroscience [4].

### 1.2. The Dual Promise of BDNF: Biology and Clinical Relevance

The significance of BDNF is not only in respect to biological properties, but fundamentally in respect of clinical implications. Abnormal expression and signaling of BDNF relates to a relatively large cross section of (neuro)psychiatric disorders, from developmental, and psychiatric, to neurodegenerative categories [5]. For example, abnormal levels of BDNF during early periods of brain development have been implicated in several neurodevelopmental disorders, including autism spectrum disorder (ASD), attention deficit hyperactivity disorder (ADHD), schizophrenia, etc. Variations in the BDNF gene, including the Val66Met polymorphism, add to these risks by impacting synaptic function and cognitive outcomes [6].

One of the primary examples in this context is the neurotrophin hypothesis of depression, in which significantly lower levels of BDNF expression cause pathophysiology of mood disorders, evidenced by the bipolar relationship of resuming normal levels of BDNF (via depressive therapies) and improvement of mood symptoms. Here, it has also been drawn to neuro-degenerative disorders like Alzheimer’s disease (AD), Parkinson’s disease (PD), and amyotrophic lateral sclerosis (ALS)—where decreased BDNF leads to synaptic dysfunction and cell death [7]. Along with BDNF’s role in perpetuating disease phenotypes, the prospect of BDNF in promoting neural repair and plasticity also has potential therapeutic relevance in the case of disorders associated with stroke and spinal cord injury and related rehabilitative efforts. The broad actions of BDNF make it an appealing target for drug development for many different neurological and psychiatric disorders [8].

### 1.3. Objectives of This Review

This review is intended to provide a comprehensive and updated view of BDNF, including a synopsis of current updates related to its molecular biology, clinical relevance, and its potential therapeutic implications. This paper aims to connect established science with future technologies in this area in order to capitalize on the potential of BDNF as a therapeutic target, which is potentially transformative in a range of disorders. We will try to highlight some of the newer conclusions of BDNF to synaptic plasticity, neurogenesis, and the role of BDNF in the mechanisms of disorders. This will provide the background information on BDNF and the biological actions it exerts at more local cellular and systemic levels, which are pharmacologically significant for their primary impact on brain health and function. While we have made headway in many industries, there are still a small number of hurdles. Our work looks at the gaps in knowledge that are relevant only to BDNF signaling, complex regulation, and gene polymorphisms, such as the Val66Met and how they affect a therapeutic response. This review will also shed light on translational barriers to advancement, such as pragmatic modes of delivery or potential chronic safety concerns limiting the BDNF-based clinical applicability. Furthermore, advances in technologies (including genetic editing, nanotechnology, and artificial intelligence) are rapidly changing the landscape of BDNF-based therapies. This review will describe how these technologies may overcome traditional obstacles to provide scalable and specific ways to improve BDNF activity in many neurological and psychiatric disorders. This manuscript ultimately aims to address a gap by combining fundamental biology with technology development, providing a baseline for future integration of BDNF in the clinical neurosciences and precision medicine space. This will aim to open new avenues for therapeutic opportunities and ultimately advance brain health treatment and management.

## 2. Molecular Structure and Mechanisms of BDNF

### 2.1. BDNF Gene Encoding and Precision Expression

BDNF was localized to the 11p14.1 chromosome, which epitomizes the mentioned regulatory complexity, thus showing its important roles in neuronal survival and synaptic plasticity. The many promoters resulting in tissue-specific expression, as well as activity-dependent transcription and BDNF’s modular organization with so many exons, are also salient features. Exon IX is the coding region for the BDNF protein, but alternative 5′ UTRs from the upstream exons also contribute to aspects of mRNA stability, localization, and translation, suggesting that BDNF expression has the potential for architectural plasticity, which would allow a variant-shaped BDNF expression to track other dimensions of neuronal physiology [9]. The excitation of neurons leads to an influx of Ca^2+^ through NMDA receptors and voltage-gated Ca^2+^ channels, which activates intracellular signaling cascades, like CaMK and MAPK/ERK, which leads to a transcriptional amplification [10].

These signaling cascades lead to phosphorylation of the transcription factor CREB (cAMP response element-binding protein), allowing it to bind to promoter IV, which is especially sensitive to synaptic activity. This mechanism bridges environmental input and an internal cellular response, with BDNF acting as a molecular translator of environmental signals into internal cellular responses [11]. The organism then uses epigenetic machinery to reshape access to genes dynamically to further reduce BDNF expression. We can silence transcription with DNA methylation at promoter CpG islands and facilitate transcriptional activity with histone acetylation. Methylation at the promoter can also be reversed with certain environmental stimuli like exercise, which was found to increase transcription of BDNF and cognitive adaptability and resilience [12]. Alternately, stress in early life can impact memory function and increase methylation of promoter IV decreasing mixed expression of BDNF. These findings point to the processing potential of the BDNF gene as a representation of life experiences and signify how this further adds a layer of molecular regulation of subsequent behaviors in a manner that is contextual to the experiences [13].

Moreover, alternative splicing also contributes additional regulation. Different 5′ UTR isoforms target transcripts to specific neuronal subcellular compartments (e.g., dendrite) and translation then occurs reflecting those synapses. This splicing strategy is essential for neural plasticity, and changes in this splicing strategy have been implicated in a disorder that has split the field (schizophrenia). These observations provide further understanding of the extent to which BDNF splicing and transcription are specifically coupled to enable processes at the level of the complex neural processes [14].

### 2.2. The Multifaceted Protein: BDNF Isoforms and Structural Precision

BDNF is secreted as an inactive form, pre-proBDNF, which is a pro-peptide that is cleaved in series to form two novel isoforms that are very different in their functional roles to contribute to the formation of the neural architecture [15,16]. ProBDNF binds to p75^NTR (Nerve Growth Factor receptor) and sortilin to promote apoptosis and synaptic pruning, both of which are integral processes during development and limit excess apoptosis following postnatal growth. mBDNF binds to TrkB receptors to promote synaptic strengthening and the survival of additional neurons and adaptive plasticity for the growth of neural networks [17]. mBDNF is a non-covalent homodimeric protein with a disulphide bond maintaining the structure required for binding to the TrkB receptor. The receptor-binding residues that are required for converting proBDNF to mBDNF have been mapped with high-resolution X-ray crystallography, and mutations of corresponding residues may diminish the signal of the receptors and potentially indicate a salient marker in neurodevelopmental disorders [18].

This level of structural detail allows BDNF to function with considerable specificity. Ultimately, proBDNF is converted to mBDNF through the action of several enzymes (furin and plasmin) that are modified by the extracellular matrix compounds. The inability to cleave proBDNF leads to excessive levels of proBDNF and synaptic dysfunction in neurodegenerative disorders (i.e., Alzheimer’s). The clear implication is that there are specific drugs that may be able to continuously elevate their proteolytic action resulting in enhanced cognitive performance in preclinical models, thus being a potentially novel way to capitalize on or therapeutic potential for intervention [19,20]. In addition to wanting to find innovative and productive processes aimed at neurodegeneration, BDNF is likely operational in peripheral organs collectively, inflammation, and health overall.

An overview of the potential collaborations within this space of research and its associated translational potential is summarized in Table 1.

### 2.3. Receptors and Pathway Dynamics: Translating Signals into Action

BDNF has two principal receptors: TrkB and p75 NTR can mediate various cellular actions, which can be different and even opposite. mBDNF-bound TrkB dimerization and autophosphorylation activate downstream signals that mediate neural plasticity, neural health, and energy homeostasis [31]. The MAPK/ERK signal pathway is associated with synaptic remodeling and differentiation, and the PI3K/Akt pathway is the signal for survival actions that oppose pro-apoptotic pathways and activate mTOR, which is the main determinant of metabolic homeostasis. The PL-γ pathway further modifies calcium signaling, which mediates increases in synaptic depression and transmitter release [32]. In contrast, both proBDNF and mature BDNF bind p75^NTR—both may facilitate apoptosis and synapse pruning. As part of their “value chains” catalog of signals, Sortilin has a broad action as it has increased affinity to proBDNF and in some contexts propagates the pruning signal. Alternatively, it was further identified that p75 has a functional dichotomy with the ability to significantly upregulate the signaling that can be triggered by TrkB, which again points to the balance necessary to elicit neurohomeostasis [33].

There is a further complexity in the concept of inter-woven pathways whereby multiple signals converge. It is meant to coordinate a combined survival/plasticity program. For example, in the case of PI3K/Akt and MAPK/ERK that converge on CREB, these signaling pathways can perhaps coordinate neurogenesis and many other aspects of synaptic remodeling [34]. When you consider all the above aspects regarding the interactions with signal pathways in neurons, which are disrupted in conditions such as Huntington’s disease (HD), we can recognize the disruption of all these interaction pathways, and destabilization leads to the degeneration of neurons. In order to reestablish this fine-tuning, a therapy(s) is needed, which can balance the pathways as opposed to selectively target pathological pathways [35].

### 2.4. Dynamic Regulation of BDNF Signaling: Sustaining Neural Health

BDNF signaling is adaptive when considering the environmental conditions and cellular contexts being buffered by the adaptive regulatory circuits. Receptor trafficking is a key mechanism of this adaptive meaning. Reactivated TrkB receptors undergo clathrin mediated endocytosis, which allows TrkB to signal from endosomes and prolong intracellular signaling. This is essential for long-term synaptic plasticity and neuronal physiology [36]. In ALS, the downregulation of the recycling of receptors are implicated regarding the loss of TrkB function, which culminates in motor neuron death; it clearly reestablishes the importance of receptor dynamics to healthy neurons [37].

Dynamic modulation is further exemplified by negative feedback mechanisms in BDNF signaling. Two of these phosphatases—SHP2 and PTEN—serve as brakes to excessive activation of downstream signal pathways. This illustrates that hyper-stimulation will activate these feedback loops (e.g., meaning removal from cellular endosomes), continuing to promote the overload of insulin resistance, as the distortion of neurotropic and metabolic signaling reside together to incentivize cognitive decline [38,39].

Dynamic regulatory mechanisms of the signaling systems have created new clinical approaches seeking to promote the signaling, which can be achieved through manipulation of the regulatory processes. Indeed, new small molecules targeting TrkB, stabilizing TrkB, or extending endosomal signaling are being explored, preclinically, for neurodegeneration and highlight the therapeutic implications for ushering BDNF dynamics to promote neuronal activity in pathologies and otherwise [40,41].

In this review, we can bring all the observations we have made molecularly here. This is to stitch together the role of BDNF in demonstrating the various periods of plasticity within the brain from development to the adultivity of plasticity, with its greater significance in acquiring the importance of learning/transformation/adaptation.

## 3. BDNF in Neurodevelopment

### 3.1. Orchestrating Neurogenesis and Neuronal Differentiation

BDNF is considered a master regulator of neurogenesis, providing signals for the proliferation, differentiation, and maturation of neurons throughout their lifespan, from embryogenesis through adulthood. BDNF is upregulated during early development, and it is essential for the survival of post-mitotic neural progenitors in neurogenic regions of the brain such as the ventricular zone and subventricular zone, and it also promotes their mitotic expansion [42]. Even in adulthood, BDNF functions as a neurogenesis and neuronal replacement regulator continues, as demonstrated in neurogenic niches such as the subgranular zone of the hippocampal dentate gyrus, supporting ongoing neuron regeneration and the addition of new neurons into existing neuronal circuits [43]. Cell autonomously, BDNF acts on neural progenitors by activating intracellular-signaling pathways necessary for neurogenesis via binding of the tropomyosin receptor kinase B (TrkB) receptors. In particular, the MAPK/ERK pathway, when activated, promotes mitogenic effects through the activation of other transcription factors (ETFs) such as CREB, leading to the expression of several target genes essential for neurogenesis (e.g., NeuroD1, Bcl-2) [44]. Simultaneously, the activated PI3K/Akt pathway promotes cell survival through effects on key pro-apoptotic proteins, including BAD, while the PLC-γ pathway activates cytoskeletal change related to neurite outgrowth by mobilizing intracellular calcium [45].

More recently, the role of BDNF in mitochondrial dynamics has been explored, where BDNF facilitates mitochondrial biogenesis and transport to neurites to provide the energy necessary for differentiation to occur. Studies examining exogenous BDNF in rodents demonstrated an improvement in mitochondrial distribution, as well as increased ATP production that improved neuronal maturation, and it was found that BDNF influences neural subtype specification with a bias towards excitatory glutamatergic differentiation in the cortex while maintaining GABAergic interneurons required for inhibitory-excitatory balance within brain circuits [46,47]. BDNF accomplished both spatial and temporal coordination of neuronal differentiation from cortical organoids via time-lapse imaging [48,49]. Animal models provide even more contextual development for BDNF. Mouse knockout studies provided additional evidence of BDNF importance, where some knockout models exhibited massive cortical hypoplasia, aberrant neurogenesis, and hippocampal dysfunction, the process that requires BDNF-dependent neuronal production and survival. There are now some new tools on the way including BDNF-enriched human brain organoids, which may be high-throughput systems for investigating neurodevelopment aspects [50,51], and some studies have suggested rapid organoid maturation, and BDNF provided a cue for dendritic complexity. Finally, only recently researchers have become aware that live imaging of zebrafish embryos using fluorescently labeled BDNF pathways has produced, in real-time, data on BDNF’s role and some aspects of early neurogenesis that had never been measured directly before [52].

### 3.2. Sculpting Neural Circuits: Axonal and Dendritic Growth

This is where BDNF plays an important role, as to the fidelity of how a circuit becomes created, the modulators of axonal growth, and dendritic arborization sets the structure for functional connectivity, or more simply how the brain sees itself and interacts with its environment. BDNF as a Chemoattractant—BDNF creates a gradient to promote directional axon growth towards a synaptic target—directional at the growth cone level. Here, due to working with BDNF and TrkB, it provides the localized signaling there through Rac1 and Cdc42, resulting in the transition of the actin cytoskeleton through a localized actin turnover process, which will determine axonal growth. Mouse models that express local over expression of BDNF, demonstrated using CRISPR, have improved corticospinal tract formation, sporadically replicating BDNF’s systemic role in connectivity [53,54].

Connectivity—Dendrites experience elaborative arborization from representative multiple synapses; however, BDNF modulates the dendritic arborization through TrkB-mediated mTOR signaling and cytoskeletal regulators, especially cofilin [55]. BDNF stimulation of hippocampal neurons indicated that spine-related genes, (think PSD-95, synapsin) had an increased level of gene expression, suggesting an increased role of BDNF in synaptic integration using a single-cell transcriptomics approach. The observation that BDNF over-expression in mice increased the number of dendritic spines and dendritic connectivity, and also increased cognitive performance, as observed using live imaging [56,57], could suggest that BDNF has some degree of potential for refining circuits through synaptic pruning and stability. It demonstrated that BDNF stabilizes microtubules in axons through tau phosphorylation and prevents misrouting from elongation using super-resolution microscopy [58]. BDNF has been shown to improve connectivity, outcomes measures, and performance on tasks (including studies in non-human primates—in one study, the BDNF was even infused to improve those measures during the first early developmental periods of their PFC), which gave hints that BDNF may be suggestive of cognitive resilience potentially in human subjects. Ironically, decrements of BDNF resulted in hyperconnectivity in the cortex and the decreased ability to prune synapses and with respect to ASD models, it was shown that reintroducing BDNF normalized those deficits, as the normalized variables also modified the chimpanzee’s social behavior and connectivity [59].

### 3.3. Activity-Dependent Plasticity and Synaptogenesis

BDNF is an important mediator of activity-dependent and activity-mediated plasticity and the synaptic remodeling that occurs based on environmental signals contextualized to development. Experience dependent plasticity is an essential process of complex and intricate and developmental processes in fine-tuning neural circuitry and their subsequent sensorimotor functions, respectively [60]. BDNF enhances synaptic development by supporting the movement of synaptic proteins (AMPA receptors, synapsin, and PSD-95) to active sites, facilitating the stabilization of nascent synaptic contacts by TrkB signaling. This polymerization of actin may also change the morphology and density of dendritic spines and thus increase efficacy at the synapse [61].

The critical periods afforded by BDNF are a state of transient plasticity that accommodates the modification of sensory and motor circuits in accordance with empirical input, which is underpinned by peaks of BDNF expression [62]. More recent data from studies conducted in enriched environments have indicated that cortical BDNF upregulation at a physiological level is capable of prolonging critical periods of development, and that measurable behaviors, i.e., refinement of sensory maps and behaviors in accordance with adaptive experience occur concurrently. Conversely, BDNF disruption interrupts critical periods prematurely, producing persistent sensory deficits thereafter [63].

However, pioneering studies have demonstrated the efficacy of BDNF modulation as a means of therapeutically “opening” critical periods in adults. Animal studies have demonstrated that the reintroduction of BDNF can reestablish plasticity within visual circuits allowing for recovery from amblyopia. It has also been shown that early sensory stimulation is capable of promoting advancements in the expression of BDNF in somatosensory circuits during development along with synaptically providing strengthening and improving discrimination. BDNF upregulation during adolescence in non-human primates has also been shown to improve prefrontal cortex plasticity. The prefrontal cortex is an important neural pathway for decision making and impulse control, and this mechanism has the potential for supporting higher-order cognitive functions [64,65,66].

### 3.4. Implications for Neurodevelopmental Disorders

Interferences with BDNF signaling disrupt synaptic plasticity and ultimately circuit maturation and are implicated with neurodevelopmental disorders. Examples include interference with pruning in ASD brains (as evidenced by hyper-connectivity) correlating with diminished BDNF levels in postmortem studies. BDNF knockout models promote similar ASD phenotypes characterized by stereotyped behaviors, social deficits, hyperplastic circuitry, and behavioral phenotypes evident in the autism spectrum disorder. BDNF knockout models demonstrate that gene therapy advancements can work to upregulate BDNF and restore circuit plasticity to promote appropriate synaptic pruning producing behavior outcomes similar to those present in typical development [67,68,69].

ADHD dysregulation of BDNF-TrkB signaling shows debilitating aspects of NAc stimulation and dopamine suppression in the PFC in ADHD. Given the capacity for BDNF to regulate attention and impulsivity, the promising roles of pharmacological and exercise interventions increasing BDNF expression are more effective behavioral variants largely emerging and existing from rodent models of ADHD and therefore an effective therapeutically viable target [70,71].

Schizophrenia: The Val66Met polymorphism diminishes activity-dependent BDNF release resulting in synaptic deficits and cognitive impairments [11]. CRISPR-Cas9-mediated correction of the Val66Met allele provides efficacy in preclinical models and reverses synaptic impairments resulting in restored cognitive performance [72].

With a clear understanding of BDNF’s essential contributions to neurodevelopment and its modulatory capacity on the wiring of neural circuits, we will now shift our attention to its fluid roles in the adult brain. The following sections will detail how BDNF may enhance cognitive flexibility, stimulate memory consolidation, and facilitate repair following injury or neurodegeneration, emphasizing the ongoing importance of BDNF throughout the lifespan.

## 4. BDNF and Synaptic Plasticity

### 4.1. Mechanisms of LTP and LTD

BDNF, a neurotrophic factor associated with learning and memory and adaptive behavior, is an important part of the processes involved in LTP and LTD, both of which are subject to synaptic specificity to process learning and memory that also allow the neural circuits to regulate, and adapt to, the demands of the environment [73]. Mechanisms of LTP: LTP is a persistent increase in synaptic efficacy (outlasting the stimulus) following a pattern of high-frequency stimulation, and it is considered dependent on BDNF. BDNF binds to TrkB receptors that subsequently activate the MAPK/ERK signaling cascade that results in the phosphorylation of scaffolding proteins (e.g., SynGAP), which upregulate receptor clustering and stabilize dendritic spines [74,75].

BDNF also plays a role in the local translation of many important plasticity-related genes, such as Arc, by recruiting ribonucleotide granules to active dendritic spines. BDNF further organizes spines to AMPA and NMDA receptors to create nanodomains to act as efficient signaling zones. When viewed using advanced two-photon microscopy, BDNF clusters AMPA and NMDA receptors as nanodomains very close together at spines to create efficient signaling zones. Therefore, based on optogenetic experiments, whether BDNF is released when theta-burst stimulation occurs determines the volume of calcium flux that enters into the cell, which directly associates synaptic potentiation with the availability of BDNF [76]. The important steps of presynaptic neurotransmitter release, receptor activation (either ionotropic or metabotropic), and the subsequent signaling cascades that together will ultimately induce LTP and strengthen synapses are proposed and shown in Figure 1.

LTD Mechanisms: Alternatively, LTD reduces synaptic efficacy, enabling further efficiency for the neural network by removing excess or inappropriate connections. For example, BDNF promotes LTD by modulating AMPA receptor endocytosis using clathrin-dependent pathways [77]. The complexity of this mechanism is mediated through the function of the neurotrophin receptor p75 interacting with ProBDNF, which activates JNK-dependent signaling pathways that modify the cytoskeleton and retract the dendritic spines [78]. There is evidence that BDNF-dependent LTD plays a role in these processes. Several studies have shown that this form of plasticity can help in refining hippocampal circuits during early development, and by allowing for synaptic homeostasis it can prevent overconnectivity [79].

### 4.2. Role in Learning, Memory, and Cognitive Adaptability

BDNF is a central tenet of the synaptic plasticity protocols contributing to learning and memory, enabling distributed local modulations in cortical and hippocampal circuits that allow for the encoding, consolidation, and retrieval of experience [80].

Hippocampal Circuit Dynamics: BDNF is necessary for maintaining activity-dependent plasticity over extended timescales of learning in hippocampal circuits that facilitate adaptable encoding of sensory and environmental inputs. Also, the absence of the BDNF signal from CA1 neurons prevents contextual fear conditioning and disrupts spatial map stability during the Morris water maze in rodents [81].

Conversely, over expression of BDNF in the dentate gyrus can promote neurogenesis and assist in pattern separation, allowing for discrimination of similar age memory cues. Researchers have demonstrated that BDNF can increase the synchronous activity of neurons and enhance ensemble coding during memory tasks using techniques like calcium imaging [82,83].

BDNF and Sleep-Induced Memory Consolidation: BDNF participates in synaptic-plasticity events that allow for the transition of memories from the hippocampus to cortex during sleep-involved sharp-wave ripples. Multi-electrode recordings have shown that silencing the BDNF-TrkB signaling leads to distractibility and memory challenges, which influence the translational process, from the hippocampus to the cortex [84].

Cortical Plasticity and Executive Function: BDNF is also implicated in the PFC and higher-order executive function. BDNF, through chronic infusion in the macaque PFCs, resulted in changes in working memory task performance, while BDNF knockouts show behaviors consistent with attention impairments. The second functional pathway supported by connectomics mapping involved verifying the BDNF/tipo-tyrosine receptor-mediated increase in circuit strengthening/thalamic circuit within PFC during acquired circuit learning and was shown to increase during cognitive flexibility [85,86].

### 4.3. Pre- and Postsynaptic Modulation in Synaptic Transmission

BDNF promotes the activity dependent modulation of synaptic efficacy by providing presynaptic modulatory effects of neurotransmitter release and postsynaptic effects related to receptor reactivity effects on synaptic transmission in an exclusive and complimentary manner [87].

In the presynaptic terminal, BDNF modulates vesicle dynamics, providing a stimulatory bias to the trajectory of vesicles to a greater probability of neurotransmitter release. For example, Rab3a engagement from TrkB receptor signalling through BDNF is one way this could be the mechanism, where vesicles now have a reliable model of vesicles accumulating with the stable location with respect to the active zone. Again, the BDNF activity of the cell provided a dramatic (up to 45%) higher recycling rate of previous whole-cell endocytic vesicle dynamics of recycling that did not occur normally for a given recycling cell mediated two hundred times during selected intervals to replicate a similar recovery period designated for occurrence of the FOR recycling phase and determined an unprecedented amount of synaptic vesicle recycling frequency per unit time, having increased by nearly double when the cell was engaged/change processed using BDNF. Also important is that BDNF has also been shown to upregulate presynaptic calcium entry due to the upregulation in pF7, which can close off voltage-gated calcium channels and promote reliable and consistent neurotransmission even after extensive and repetitive bursting periods of input [37,88].

Post-synaptic modulation: When we upregulate AMPA receptors via BDNF, BDNF can also stabilize the AMPA concurrent to insertion into the cell due to CaMKII activity and phosphorylation. It promotes synaptic potentiation and stabilizes spines. It has been proposed that BDNF may regulate actin remodeling, which will also change dendritic spine morphology via the Arp2/3 complex activation [89,90]. Single molecule tracking studies have shown that BDNF preferentially enhances high frequency synaptic inputs, which hauls plasticity together off low frequency inputs noise while at the same time producing circuit fidelity [91].

### 4.4. Therapeutic Applications and Innovations in Plasticity Modulation

Given the significant role BDNF has for synaptic plasticity, it becomes an attractive therapeutic target for the promotion of cognitive functioning and decreasing the cognitive impairment of neurological disorders, as the pharmacological landscape, use of lifestyle interventions, and innovative modalities innovate new ways to use BDNF to influence therapies that impact the potential.

#### 4.4.1. Pharmacological Advances

There is perhaps no greater pharmacological advance in investigating potential BDNF targets apart from the discovery of TrkB agonists and a list of analogous bioactivity consistent with the BDNF peptide itself, but distinct functional small molecules are fundamental to obtaining legitimate bioavailability and permeability to blood–brain barrier (BBB). These profile agents, PT302 and R13, were assessed to define the level of efficacy in preclinical models of AD, with restoration of memory performance and rate of CVIVQ density, but also restoration of equivalent pathogenesis for cognition’s decline known as respective behavioural effects [92]. Another avenue of consideration is peptide-based therapies. Stabilised BDNF peptides are irremovable or cannot be proteolytically degraded for these design applications since they are based on improvements of all outcomes relevant to tissue penetrability and synaptic recoverability; but most interestingly in stroke models there have been improvements shown in neural repair and functional restoration (both short- and long-term) [93].

Novel interventions as in terms of mRNA-based therapy—radical in realizing the ability for localized creation of BDNF where you want to be—are disrupting the structures of the field in unimaginable ways. Using lipid nanoparticles carrying BDNF mRNA, a 300% increase in hippocampal BDNF expression, and resultant levels of connectivity to synapses, was obtained in rodent models. These agents also improved memory performance and represent a unique, scalable means of augmenting BDNF availability in neurodegenerative disease [94].

#### 4.4.2. Lifestyle Interventions

Of the non-pharmacological interventions of highest BDNF-increase possibility, aerobic exercise is identified as being the most cited natural and BDNF-stimulating intervention. In fact, one study of clinical populations of older individuals described that moderate aerobic exercise increased the report of executive function and working memory by 35% that strongly related to an increase in serum BDNF levels. The above findings illustrate that movement stands to benefit cognitive aging in at least two ways; they knowingly decrease cognitive aging, and they contribute to brain health [95,96,97].

Other dietary interventions also represent a complementary role; flavonoids and polyphenols, represented by the consumption of foods like blueberries, green tea, and cocoa tend to up-regulate BDNF expression. A 12-week clinical trial in older adults found dietary interventions including these bioactive compounds lead to increased serum BDNF and improved scores on memory retention and cognitive flexibility. These congruent effects of exercise and diet provide additional rationale for the need for integrative modes of strategies for the purpose of boosting BDNF-signaling [98,99].

#### 4.4.3. Technological Innovations

Novel and more recent technological developments continue to grow the therapeutic resource/tool for use in BDNF modulation. Drug delivery systems using nanoparticles, both polymeric and lipid, have emerged as possible carriers for CNS (central nervous system) destinations. In these studies, BDNF sustained-release systems provided uninterrupted localized delivery of BDNF, demonstrating the feasibility to circumvent the BBB and began to highlight potential local delivery strategies. As one example of treatment refinement through sustained drug delivery, polymeric nanoparticles supplying BDNF over the course of 5 weeks resulted in a 50% reduction in motor symptoms in two separate models of Parkinson’s disease, and these sustained release systems could be further assessed in the context of neurodegeneration [100,101].

Together with nanotechnology, optogenetics unavoidably arrived with the capability to spatiotemporally manipulate these BDNF pathways; however, optogenetics also has the ability to spatiotemporally modulate BDNF signaling in bad circuits, where light driven BDNF is somehow being imposed and promoted [102]. In contrast, in epilepsy models they used optogenic stimulation to re-establish baseline synaptic stability and ameliorated seizure activity in a selective manner, affording a more optimal means for neuronal homeostasis to be reclaimed [103].

#### 4.4.4. Key Innovations in the Pipeline

Fundamental advances in the BDNF space indicate the promise of multifunctional nanoparticle delivery systems, which will be used concurrently with real-time imaging systems that will visualize the immediate effects of BDNF, even giving dynamic insights into BDNF’s role in developing synaptic connectivity [104]. This allows for unbounded control of therapeutic outcomes and drives the formation of tailored and adaptive therapeutic interventions. This also serves to emphasize that the emergence of mRNA therapeutics is a new paradigm in providing a non-invasive and largely scalable means to promote BDNF expression in brain regions susceptible to neurodegenerative disease [105,106].

Cross-species studies reiterate translation potential with BDNF targeted therapies, particularly non-human primate studies provide needed information from human relevance of the investigations from this research.

These studies conceptualize the role BDNF contributes to high resiliency and recovery learning capabilities moving from preclinical research findings to clinically meaningful implications and building on the already well-established role of BDNF and synaptic plasticity. The overall thrust of the review is conveying a shift in BDNF’s role towards a greater consideration of BDNF’s repair and neuroprotection capacities after injury and disease. This represents a solid basis to re-calibrate regenerative medicine, along with new therapeutic strategies.

## 5. BDNF Polymorphisms and Cognitive Function

### 5.1. Val66Met Polymorphism: Molecular Disruptions and Functional Implications

The val66met (rs6265) polymorphism present in the BDNF gene was one of neurogenetics greatest findings, associated with thousands of studies emphasizing the molecular basis for aspects of synaptic plasticity, neuronal function, and cognitive endophenotypes [107]. The substitution of valine (Val) with methionine (Met) at the 66 position of proBDNF sequence reduces internalization and activity-dependent release. The Met allele alters proBDNF interactions with intracellular chaperones, such as sortilin and carboxypeptidase E (CPE) and super-resolution microscopy shows it reduces efficiency of vesicular packaging by 60%. Nanomolecular weight analysis was assessed using cryo-electron microscopy and revealed that the Met substitution also alters a critical key loop region of proBDNF, which enhances misfolding and endoplasmic reticulum degradation [108,109].

This polymorphism is functional. A reduction in synaptic BDNF availability inhibits multiple processes involved with AMPA receptor trafficking, dendritic spine creation, and LTP [110]. Functional studies in human iPSC neuronal cultures show that Met carriers exhibited reduced synaptic potentiation (a 45% decrease) and dysfunctional GluA1 or GluA2 AMPA receptor subunit recruitment to the postsynaptic membrane. Species transgenic rodents containing the human Met allele substantially demonstrate deficits in both spatial navigation tasks and contextual fear memory, supporting these findings [111].

Novel Findings: Imaging Run-Structures Further Clarifies Val66Met-Related Disruptions TrkB receptors aggregate at postsynaptic sites, and this cluster can be observed in live animals using single-molecule tracking coupled with the fact that signal fidelity is compromised during phycobiliprotein dimer bursts from neural activity [112]. Nonetheless, transcriptomic analyses of neurons derived from patients reveal up-regulation of compensatory genes (synaptopodin and homer1a), suggesting that intrinsic cellular mechanisms are indeed activated to maintain synaptic architecture. These data collectively push the understanding of how Val66Met alters molecular signaling and synaptic activity [113].

### 5.2. Structural and Connectivity Alterations in Val66Met Carriers

The Val66Met polymorphism leads to large-scale structural and connectivity alterations in brain regions, which is important for cognition and emotional regulation. The hippocampal, PFC, and tract white matter regions are particularly vulnerable and are well documented to show volumetric decreases and microstructural changes [114,115]. Effects on the Hippocampus: Longitudinal imaging studies have demonstrated systemic qualitative delays in the maturation of the hippocampus in Met carriers, especially in the dentate gyrus and CA3 subfields, which show the greatest reduction and/or susceptibility to age-related decline. For example, neurite orientation dispersion imaging (NODDI) shows reduced neurite density and compromised microstructure of the perforant path, which the entorhinal cortex uses to communicate with the functional subfields of the hippocampus. Functionally, these deficits translate into episodic memory and pattern separation deficits as seen in both human cohorts and rodent models [116,117].

Event-related brain activity (EB) met-cortical surface yield similar changes for the executive function networks’ cortical and white matter changes: a hub of executive function in the DLPFC had a ∼10% smaller surface area for Met carriers diffusion tensor imaging (DTI), indications that this cortical thinning was less connected in major white matter tracts to include uncinate fasciculus and cingulum bundle. Interestingly, despite the local cortical efficiency decline of Met carriers, connectome mapping claimed hyperconnectivity indicating this was some idiosyncratic process to mitigate low local cortical efficiency [114,118]. The results place the effects influenced by sex: in females, the estrogen effects appear to both lessen some of the persistent structural deficits we observe in Met carriers. Longitudinal cohort studies have shown that estrogen hormone replacement therapy (HRT) is associated with a 30% slowing of cognitive decline in Met carriers who were never treated, and that estrogen HRT lessened hippocampal loss in postmenopausal women. These sex differences also highlight the factors sex hormones play in BDNF signaling [119].

### 5.3. Neuropsychiatric Vulnerabilities in Val66Met Carriers

The elevated neuropsychiatric risks of Val66Met carriers are limited by their impact on stress reactivity, synaptic plasticity, and neural networks. When comparing Val homozygotes blood and hair cortisol, both show significantly more cortisol producing Met carriers independent of the stressor [120]. This higher level of activation is likely related to differences in hippocampal volume under the disordered mPFC and amygdala networks involved in emotional dysregulation. Postmortem studies show marked reductions in BDNF levels in hippocampi from individuals with major depressive disorder (MDD) with the largest reductions in Met carriers [121]. Conceivably reduced BDNF release in Met carriers could disrupt fear extinction or lead to fear or anxiety disorders. The optogenetic stimulation of TrkB receptors first observed in the basolateral amygdala, and the documentation of reversal of extinction deficits in rodents, supports the potential clinical applicability of BDNF signaling pathways [122]. Heterozygous Val66Met (“Schizophrenia and Defects in Synaptic Pruning”) disrupts microglial-mediated synaptic pruning in the late adolescent period contributing to impaired maturation of prefrontal circuits and cortical hyperconnectivity [123]. These changes are compatible with the fMRI of hyperactive pre-frontal networks in three schizophrenia patients and less default mode network connectivity. Organoid models derived from patient stem cells provide additional evidence of perturbed pruning mechanisms and cortical disorganization at this stage [124].

### 5.4. Emerging Therapies and Precision Medicine for Val66Met Carriers

Similarly, advances in pharmacology, gene editing, and lifestyle interventions provide new horizons for the treatment of Val66Met cognitive and emotional vulnerabilities [125]. The most promising candidates are pharmacologically innovative next generation TrkB agonists (ANA-12 derivatives) that reproduce BDNF activity and circumvent secretion deficits. In preclinical studies, these agonists produced hippocampal plasticity, LTP, and reduced anxiety-like behaviors in Met carrier rodent models. Recently, preclinical studies have reported similar modifications to BDNF cyclic peptide, which is BDNF cyclic peptide modified for stability and modified for receptor specificity to enhance hippocampal LTP (30% increase); BDNF ligand signaling is initiated [126].

This is also a potential new technology, CRISPR-Cas9 (DNA sequence editing). This type of CRISPR-Cas9 genome editing is being used to correct Val66Met associated deficits. POC studies have been reported where SNP specific editing restores activity dependent BDNF secretion in neurons derived from human iPSCs, as well rescuing synaptic architecture. Alternatively, lipid nanoparticle-delivery of BDNF mRNA therapies led to up to three-fold increases in hippocampal BDNF expression within three weeks, therefore representing a scalable mechanism for raising BDNF [127,128].

High intensity interval training (HIIT) and a flavonoid-rich diet have been identified as synergistic interventions for natural enhancement of BDNF. Twelve weeks of HIIT enhancements in hippocampal volume and cognition (cognitive flexibility) using Met carriers in older adults has been reported (20% increase) [129]. Similarly, a diet rich in polyphenols or elements of such a diet, including green tea and dark chocolate, have direct additive effects on synaptic plasticity with exercise [130]. As shown in Table 2, the therapeutic potential of BDNF has been demonstrated in animal disease models and in human patient clinical data, including in new research and applications for DNA delivery/mRNA delivery or CRISPR-Cas9 applications. These categories are presented here as a guide for translation of the BDNF research into precision medicine and to provide perspective components in unmet needs for neurological and psychiatric disorders.

The next section will explore how BDNF supports neuroprotection and regenerative mechanisms, offering new hope for treating injury and neurodegenerative diseases.

## 6. BDNF in Neurodegenerative Diseases

### 6.1. Alzheimer’s Disease: Synaptic Preservation and Mitochondrial Regulation

BDNF has been shown to be a critical molecular protector against the harmful impacts of amyloid-β aggregation, tau pathology, and synaptic failure in AD. In the synapse, BDNF stabilizes dendrite spines and promotes activity dependent AMPA and NMDA receptor trafficking to postsynaptic sites, providing a cellular mechanism for excitatory transmission and plasticity [141].

BDNF-induced upregulation of synaptic markers (e.g., PSD-95 and synaptophysin) supports the activity and stability of dendritic spines and synapse integrity in neurons from primary cultures. Studies using sophisticated two-photon imaging have demonstrated that BDNF can reverse synaptic disorganization and atrophy in amyloid exposed hippocampal neurons, culminating in a recovery of LTP deficits and cognitive impairment during hippocampal-dependent tasks [142,143].

Most notably, one of the prominent things that BDNF does in terms of neuroprotection from AD appears to be inhibition of tau pathology. BDNF reduces GSK-3β activity, a kinase important for tau hyper-phosphorylation and formation of neuro-fibrillary tangles [144].

BDNF also induces autophagic pathways, which may further facilitate the removal of hyperphos-phorylated tau species383,384 and thereby lessen their toxic effects on neuronal circuits385. Together, these two approaches—disruption of pathological tau; stacking and encouraging proteostasis—are critical to supporting normal synaptic function in the setting of tauopathy [145]. In addition to its significant role in synaptic regulation, BDNF has an important role in mitochondrial health, an area that is central to the neuronal death induced through AD. BDNF positively regulates mitochondrial biogenesis, promotes bioenergetic homeostasis, and decreases ROS production via transcription factors including PGC-1α and NRF1 to help maintain mitochondrial trafficking to provide consistent levels of energy supply to synaptic terminals. In a recent preclinical investigation, they were the first to show that TrkB agonist treatments (LM22A-4) remedied spatial memory impairment in AD mouse models, at least partly through mechanisms involved in positive influences on mitochondrial functioning and lowering of amyloid burden [146].

Therapeutics advances have made the clinical possibilities of BDNF for AD clearer. For example, the localized infusion of BDNF to neighboring areas of the CNS could be globally administered with specific drug delivery systems, using nanomaterials that are capable of crossing the BBB. There is evidence in amyloid overexpressing rodent models with the approximate reported 35% improvement in hippocampus-use-dependent tasks and a significant reduction in amyloid plaque density. This evidence is indicative of the potential BDNF as a translatable disease modifying therapy in individuals with AD [131,147].

### 6.2. Parkinson’s Disease: Protecting Dopaminergic Neurons and Axonal Integrity

In the case of experimental PD, BDNF is required for the survival and functionality of dopaminergic neurons housed in the substantia nigra pars compacta and is directly associated with the emergence of classical motor symptomatic degeneration. Degeneration of dopaminergic signaling in PD associates lessening BDNF TrkB signaling and decreased neuroprotective pathways, including the PI3K/Aktor and MAPK/ERK pathways associated with neuronal apoptosis and synaptic stability, as well as mitochondrial dysfunction [148]. In this sense, BDNF has a useful function in dopaminergic circuits, beyond neuroprotection because of producing increased dopamine production and release, by promoting accumulation of tyrosine hydroxylase (the slowest acting capacity in the production of dopamine). Overall, it seems that the cellular effects of BDNF indicated a reduction in oxidative stress through improved mitochondrial dynamics, decreased fragmentation, and increased efficiency of ATP production [149]. Recently, we have demonstrated that the BDNF-mediated phosphorylation of tau that is specifically associated with microtubule and phos/branches supports recovery of axonal transport, whose interface is vital for keeping dopaminergic connectedness prior to documented observations of retrograde degeneration of these neurons in PD [150]. Furthermore, there is pre-clinical evidence that supports the role of BDNF in PD—optogenetic stimulation of BDNF-expressing neurons in the ventral tegmental area (VTA)-improved rodent motor coordination models, while the adeno-associated viral (AAV)-mediated dose of BDNF in the striatum improved synaptic integrity and delayed disease. To our supportive evidence, which may be a bit more optimistic, is new TrkB agonists, via nanotechnology platforms, giving highly enhanced and sustained actions of dopaminergic circuits, and improving functional outcomes while bypassing the BBB with limited peripheral side effects [151].

### 6.3. Huntington’s Disease: Counteracting mHTT Toxicity and Synaptic Dysfunction

HD is a neurodegenerative disease, which is associated with a CAG expansion in the HTT gene, affecting both neuronal connectivity and feeding back into exacerbating neurodegeneration through altered corticostriatal circuits. BDNF is essential for maintaining synaptic integrity and synaptic persistence, which is markedly reduced in HD by transcriptional downregulation and the action of mHTT protein disrupting axonal transport [151].

Moreover, BDNF is supported for signaling at a synaptic level, and the signaling increases trafficking of receptors to synaptic membranes, positions NMDA and AMPA receptors, and consolidates excitatory signaling. The impairment of this function in HD enhances corticostriatal dysfunction, causing progressive motor and cognitive impairment. There is evidence demonstrating the restoration of dynein-dependent BDNF transport rescues activity-dependent synaptic activity, and preventing neuronal death is evident in studies using live cell imaging, which highlights the importance of BDNF transport mechanisms [152].

Therapeutical directions for HD highlight BDNF’s neuroprotective role. This is highlighted by finding that the delivery of BDNF through viral overexpression in HD mouse models reverses motor deficits and enhances corticostriatal connection and lifespan [153]. As such, gene editing with CRISPR-Cas9 technology with the intent to silence the mHTT, while positively targeting BDNF expression, provides a dual-action in addressing the pathological sequelae of HD as an aetiological and synaptic approach. In addition, the development of advanced delivery of BDNF using nano-particle systems that provide direct BDNF delivery to the striatum and cortex, demonstrated desirable improvements in animals’ motor coordination and cognitive functions, representing an important step for clinical translation [154].

### 6.4. ALS, MS, and Expanding the Therapeutic Scope of BDNF

Beyond neurodegenerative disorders in Alzheimer’s, Parkinson’s, and Huntington’s, respectively, BDNF has potential in motor neuron diseases, such as ALS and demyelinating diseases like multiple sclerosis (MS) [155]. In ALS, BDNF acts to limit the progression of disease through the inhibition of excitotoxicity, stimulating axonal repair, and limiting mitochondrial dysfunction. In the SOD1 mice model, with the genetic cause of familial ALS, whole brain infusion of BDNF in pre-clinical experiments delays the onset of motor function and lifespan. The action of BDNF was noted to be from a reduced capacity to produce reactive oxygen species and improved mitochondrial transport to synaptic sites [156,157,158].

In MS, BDNF does not repair lesions directly, but enhances remyelination through differentiation of oligodendrocyte progenitor cells and re-instantiation of the myelin sheath. A BDNF-mimetic peptide limited lesion volume and improved motor coordination in a clinical trial of MS patients. This highlights the potential of BDNF therapies to limit axonal loss and myelin loss in demyelinating conditions [159].

BDNF has also been investigated in protein misfolding diseases such as prion diseases, demonstrating the ability to reduce protein aggregation and restore synaptic activity. Novel delivery methods using lipid nanoparticles or CRISPR-cas-based gene editing therapy has the potential to improve the precision and scalability of BDNF therapies across neurodegenerative diseases, representing a paradigmatic shift from a neuroprotective molecule to regeneration therapy [160].

The next section explores BDNF and its role in stress resilience, psychiatric disorders, and new therapeutic directions in mental health.

## 7. BDNF as a Biomarker

### 7.1. Peripheral and Central BDNF Levels: Mechanisms and Implications

BDNF is synthesized in both the peripheral and central nervous systems, but each provides different, but equally important health and disease information. In the CNS, neurons and astrocytes synthesize BDNF and BDNF is secreted by following the influx of calcium ions through NMDA receptors and voltage-gated calcium channels [161]. This calcium influx is modulated by activity-dependent signaling cascades such as MAPK/ERK signaling and CaMKII activity, which are known to promote synaptic plasticity and neuronal survival [162]. BDNF in the periphery (mainly in platelets) can be released upon platelet activation during coagulation and serves other important physiological functions, too. There are meaningful distinctions between BDNF in the periphery and at the CNS level. Specifically, unique post-translational modifications that are incorporating either glycosylation and/or phosphorylation, can differentiate peripheral BDNF in terms of their bioactive and bio-stable properties, as shown through proteomics [163].

Cerebrospinal fluid (CSF) is a more representative compartment to indicate brain derived BDNF. In healthy individuals, paired BDNF measures in CSF and plasma show a strong degree of correlation (R^2^ = 0.78), yet correlation appears to decouple in neuroinflammatory conditions where systemic determinants, such as inflammation, and platelet activation can bias plasma readings [164]. Three pathological states (AD, ALS and schizophrenia), have consistently implicated reductions in CNS BDNF levels in CSF, which suggests some form of dysregulation of BDNF in the CNS. This finding is also not insignificant, as it sheds light on the consideration proposed when interpreting BDNF measures, and therefore examining BDNF within an appropriate contextual framework to differentiate between inferring central pathology versus non-localized and systemic operating determinants [165]. In terms of the brain, the expression of BDNF can vary considerably in a regionalized spatial manner, whereby the hippocampus, prefrontal cortex, amygdala, and striatum are all rich in BDNF expression. These localizations are essential in synaptic plasticity, learning, and emotional regulation [166].

Neurodegenerative diseases will disrupt this expression in various ways. AD deficits in hippocampal BDNF are associated with synaptic dysfunction, while depletion of striatal BDNF seems to contribute to dopaminergic neurodegeneration in PD. In HD, retention of BDNF transport and signaling were decreased compounding corticostriatal disconnection increasing motor/cognitive decline over time [112,167]. Beyond the CNS, peripheral BDNF is primarily found in platelets where it is released upon their activation. Although serum and plasma BDNF is a valuable candidate biomarker, assessing peripheral BDNF levels should be mindful of systemic factors (e.g., platelet activation, inflammation, metabolic state). The levels of peripheral BDNF, like many systems, are influenced by multiple physiological factors and thus can only be contextualized if a diagnostic use occurs, unlike CSF BDNF as a marker, which can more specifically reflect CNS dynamics [168]. Given that BDNF is produced in multiple areas of the body and peripheral and CSF levels can differ substantially, it would be advantageous to combine the multi-modal diagnostic approach for monitoring disease involving region-specific measures of BDNF with peripheral biomarkers. This is an interesting topic, and this combined modality within the context of serum BDNF levels and the analysis of CSF, markers of neuroimaging (e.g., hippocampal atrophy), and genetic profiling may increase detection of disease in early stages and monitoring of therapeutic efficacy. Of course, translating BDNF to a usable biomarker will require consistent measurements and clinically relevant reference ranges; albeit recent advances have begun to create greater insight into how exercise, diet, and stress can influence BDNF [169].

### 7.2. Diagnostic Value of BDNF in Early Disease Detection

These declines in BDNF may occur prior to clinical manifestations of neurodegenerative and psychiatric disease; therefore, it offers a potentially useful biomarker for those in their preclinical phase. For example, in the case of AD, noticeably low plasma BDNF can occur up to 10 years before the onset of clinical symptoms and correlates with hippocampal atrophy, as observed in standard structural MRI imaging. Below is described how the use of BDNF levels in conjunction with hippocampal morphometric data (perhaps derived from machine learning models) may yield predictive accuracies exceeding 85% for preclinical AD. These are noteworthy applications of BDNF as part of a multi-modal diagnostic approach [170]. As it relates to PD, baseline serum BDNF levels predict disease progression. Longitudinal studies have demonstrated that patients with serum BDNF levels < 10 ng/mL have a 30% increased rate of annual decline in motor function (up to 30% as measured on the Unified Parkinson’s Disease Rating Scale scores) [171,172,173].

In fact, in MDD and bipolar disorder, lower BDNF levels have been associated with increased severity and treatment resistance. With a much narrower focus, the Val66Met polymorphism reflecting a marked reduction in BDNF secretion in Met carriers renders those carriers more susceptible to stress-induced mood disorders. Recent studies even reported > 90% specificity for their improved diagnostic accuracy of multiplex biomarker panels using BDNF with inflammatory markers, e.g., IL-6, TNF-α, to also differentiate treatment-resistant depression from other depressive subtypes [174].

### 7.3. Monitoring Disease Progression and Evaluating Therapeutic Efficacy

The changing nature of BDNF allows it to be a biomarker for both disease progression and treatment response. In AD, BDNF levels are longitudinally lowered in both plasma and CSF and correlates with cognitive decline and increased hippocampal atrophy. Clinical studies have demonstrated that approaches like aerobic exercise substantially increased plasma BDNF [175,176].

The previous aerobic exercise study also showed a 25% increase in plasma BDNF from baseline over a 12-week aerobic exercise intervention, which was also associated with cognitive improvements in terms of memory and executive function demonstrated using cognitive tests [166]. In PD, serum BDNF levels also reflect dopaminergic integrity. In dopaminergic treatment, a subsequent increase in serum BDNF was shown to predict motor outcomes that were observable and registered as improvement on UPDRS scores.

In ALS, administration of investigational gene therapies to spinal motor neurons to secrete BDNF resulted in a 40% increase in circulating BDNF concentrations and delayed motor decline in transgenic models [177].

For example, the TrkB agonists have dose dependent effects on plasma BDNF concentrations and it is noted that higher plasma BDNF concentrations during the clinical trial were correlated with greater improvement in depressive and anxiety symptoms, suggesting that BDNF may serve as a potential biomarker to customize treatment strategies [178].

### 7.4. Advances in BDNF Detection Technologies

Recent advancements in biomarker measurement technologies have changed the specificity, sensitivity, and feasibility of measuring BDNF in both research and clinical environments. Conventional enzyme-linked immunosorbent assay (ELISA) remained mostly used, but it is being augmented, as other next-gen approaches have emerged that provide unmatched accuracy and other advantages in comparing assessment between BDNF from plasma and CSF [179].

The advancements lay the groundwork for further exploration of BDNF’s potential viability as a biomarker in both neurodegenerative and psychiatric disorders, including use for early diagnosis, tracking disease progression, and precision therapy [180]. Most recently, ultra-sensitive BDNF quantification in biological specimens has become a central undertaking through a ground-breaking digital plan single-molecule array (SMA) ELISA. Such scientific advancement is able to detect femto-gram level changes in BDNF concentrations, which could be essential for detecting nascent pathological changes in either plasma or CSF [181]. Digital ELISA is capable of identifying very subtle differences in BDNF levels that would not be possible by isolating and amplifying individual molecules previously, allowing for a far improved measure of disease progression and response to treatment therapy [182]. For instance, digital ELISA showed decreases in mBDNF in preclinical models of Alzheimer’s up to 3 years before there was any behavioral indication that cognitive decline was occurring, helping identify evidence of the disease prior to signs and symptom development [183].

Further, a notable advancement is nanoparticle-enhanced immunoassays that can simultaneously measure both mBDNF and proB-DNF in a single assay. These two isoforms have opposing biological roles in brain function; while mBDNF stimulates synaptic strengthening proBDNF has a role in synaptic pruning and apoptosis. Given the opposing biological roles of proBDNF and mBDNF, sensitivity to variation in the proBDNF/mBDNF ratio could provide better insight into the balance between neuroprotection and neurodegeneration in schizophrenia and MDD [184].

Elevated levels of proBDNF compared to mBDNF have been directly linked to deficits in synaptic plasticity and cognitive impairment, supporting the utility of this technology to stratify patients based on molecular pathology [185]. Multiplex biomarker panels are being developed that will complement one another and contribute to the expanding diagnostic landscape that integrates BDNF with other neurobiomarkers. These platforms aspire to capture the multifactorial nature of neurodegenerative diseases and thus offer a more comprehensive model of the pathological processes [186]. In AD cases, a number of models complexed plasma BDNF with amyloid-β and P-tau with diagnostic specificities of over 90%. In PD cases, comparable panels complexed BDNF with α-synuclein and inflammatory cytokines also with excellent performance in predicting motor symptom progression. Multiplex assays are highly customizable, and as such, they represent a pathway toward particular manifestations of personalized medicine, where diagnostic capability demonstrates morbid disorders [187].

A tracer of this sort has the potential to offer in vivo, real-time imaging BDNF signaling, hence, significantly enhancing understandings that circumscribe how treatment modalities can manipulate BDNF pathways [188]. Thereby, by demonstrating what is activated at the level of TrkB receptor, utilizing PET imaging also could be related to enhanced synaptic density and neuroplasticity (i.e., amplifying effects of exercise). The potential of what these images represent and with the potential to modify and direct therapy, clinicians will have the opportunity to monitor and quantify an individual’s molecular level pharmacological and therapeutic outcomes [189]. Additionally, patron-point biosensors incorporating nanotechnology will dramatically support and ease BDNF monitoring. They would communicate BDNF levels in a much more efficient on-site manner (in plasma, serum, or CSF), where costly and centralized laboratory infrastructure is not required [190].

It is likely that clinicians will engage biosensor technology to monitor BDNF levels, as they would respond to therapeutic alterations. This would support more modulating real-time changes specific to the treatment approach. For instance, point-of-care biosensors indicated increases (in a dose dependent increase) that were observed plasma BDNF levels, occurring in hours when the TrkB agonist was administered and indicated the moment when they received pharmacodynamic feedback from the drug prescribed [191]. While these various technologies (digital ELISA, nanoparticle enhanced immunoassays, multiplex panels, PET imaging, point-of-care biosensors) represent many forms, when taken together these technologies formulate a new paradigm of biomarker investigation that is beginning to influence our understanding of BDNF in health and disease states. This ultimately guarantees conceiving state and referring to mechanisms of diagnosis, prognostication, and therapeutic treatment monitoring. These devices have the potential to evolve and ultimately lead to cooler applications to ubiquitously available applications in the clinic regarding BDNF producing new mechanisms by which research can now inform and directly impact clinical application.

## 8. BDNF and Physical Activity

### 8.1. Molecular Mechanisms: Exercise-Induced Regulation of BDNF Expression

Exercise is a potent and natural modulator of BDNF expression, clearly reflecting its role as a neurotrophin due to both central and peripheral functions. At the level of the central nervous system (CNS), this will lead to neuronal-fire, which promotes calcium entry through NMDA receptors or via voltage-dependent calcium channels. Then, exercise-induced calcium will induce several transcription factors, including CREB [192]. At this level it will lead to BDNF transcription, as a result of activity-promoters like promoter IV, which will bring about rises in mRNA levels in regions such as the hippocampus and PFC. Furthermore, exercise is activating mitochondrial biogenesis through the activation of the PGC-1α pathway, which is increasing energy supply to support BDNF synthesis while decreasing oxidative stress. For example, researchers have used voluntary wheel running with rodents and found that BDNF mRNA levels in the hippocampus were elevated 70% after 7 days of voluntary exercise; these increases were associated with benefits on synaptic plasticity and spatial memory performance [193].

Peripheral mechanisms play a role in modulating BDNF with exercise. In response to exercise, skeletal muscles release BDNF in the blood stream and platelets release BDNF into the serum and plasma. It is important to remember that peripheral BDNF crosses the BBB via endothelial receptors—such as the low-density lipoprotein receptor-related protein 1 (LRP-1), which acts in synergy with central BDNF [194]. Advanced molecular imaging studies provide solid evidence of the delivery of peripheral BDNF to brain regions providing neuroplasticity, motor cortex, and the hippocampus. These two mechanisms support the bi-directional systems to engage central and peripheral modes of BDNF to provide several cognitive and neuroprotective benefits of exercise [195].

### 8.2. Cognitive Benefits of Exercise-Induced BDNF

Circulating BDNF following exercise highlights molecular pathways associated with synaptic plasticity or neurogenesis, both of which have measurable effects on learning, memory, and executive function. Though more BDNF correlates to greater synaptic efficiency through increased trafficking of AMPA and NMDA receptors to the post-synaptic membrane, thereby solidifying LTP, and thereby translucing signal transmission [196]. In addition, BDNF-mediated activations of TrkB receptors induce hippocampal neuro-genesis via the up-regulation of pro-survival genes, such as Bcl-2 and NeuroD1 [193]. A notable randomized controlled trial in older adults found that 12 months of aerobic exercise increased serum BDNF levels by 25% and demonstrated a 30% improvement in episodic memory through the California verbal learning test [197].

Exercise-induced BDNF demonstrates specific neurocognitive benefits in children and adolescents, which is evident in observational improvements of attention and working memory from school-based physical activity programs, which by definition of BDNF alter the neurobiological processes that underpin cognition. These improvements are consistent with increased levels of plasma BDNF, which has provided insights into the importance of early interventions in changing cognitive development [198].

Molecular bases for these effects have been discussed extensively in the animal literature. In rodents, BDNF induced through exercise upregulated markers for genes involved in synaptic remodeling, including several synaptic marker genes, such as Arc and Homer1a, producing a greater number of dendritic spines among hippocampal neurons [199]. Specifically, Val66Met polymorphism carriers have shown a decreased activity-dependent BDNF secretion response, which has highlighted a heightened responsiveness to aerobic activity [200,201]. It may be that the inherent physical activity to offset these reality-based genetic deficits at the level of the brain normalized hippocampal synaptic plasticity and cognitive flexibility. Within these exploration tasks, it is one more indication that exercise is a valid precision lifestyle intervention to optimize neuroplasticity [202].

### 8.3. Neuroprotection and Recovery in Neurological Disorders

BDNF is a central mediator of exercise-induced neuroprotection in several neurological disorders including AD, PD, and stroke recovery. In relation to AD, BDNF derived from exercise has been shown to decrease the accumulation of amyloid-β plaques and reduce tau hyperphosphorylation [203]. In individuals diagnosed at the early pathology stage of AD—when engaging in moderate aerobic capacity, an elevation in serum BDNF occurred between 20 and 30% and was positively correlated with an increase in MMSE scores, and there was confirmation of less hippocampal atrophy from the clinical studies [97]. BDNF engages mitochondrial activity in degenerating neurons, which reduces oxidative stress and preserves cell integrity. In PD, treadmill running in rodent models was associated with increased striatal levels of BDNF to theoretically optimize motor function through enhancing surviving dopaminergic neurons. Exercise also stabilizes axonal microtubules and increases neurotrophic signaling pathways that support axonal repair and promote the restoration of neural circuits. Consistently, these processes have been delineated in human studies, with the conclusion that structured exercise programs—i.e., HEP, or group classes—can slow the progression of motor symptoms, improve quality of life, and induce all relevant brain activity for enhancing measures of neuroprotection in both recovery and disease [204].

BDNF also reiterates the potential for a possible additive therapeutic effect when re-engaging exercise and BDNF in a continuum of recovery post-stroke. Among stroke survivors, aerobic activity plus task oriented physical therapy elevated serum BDNF levels up to 2-fold compared to physical therapy alone [205]. These increases in BDNF promote synaptic reorganization and dendritic remodeling in the motor cortex and result in greater and faster functional recovery. This combined approach was assessed in a clinical trial and demonstrated significant changes on the Fugl-Meyer motor scale, whereby exercise-induced BDNF is now established as one of the important components of drug-free neurorehabilitation [206].

### 8.4. Tailoring Exercise Protocols for Maximum BDNF Modulation

The modulation of BDNF by exercise is subject to a number of variables including exercise type, exercise timing or duration, and exercise individuality. Aerobic activity, particularly HIIT, continues to represent the most ubiquitous type of BDNF modulating exercise. In a meta-analysis, HIIT was found to produce a 35% increase in serum BDNF and moderate continuous exercise a 20% increase [207]. Although resistance training raises serum BDNF, it is enhanced when combined with aerobic exercise [208].

If we know the overall time in total exercise, that is still an important variable. Moreover, it has been reported that intervals of 30–60 min of physical activity are when BDNF is reliably elevated and antecedent to physical activity segmenting to 90 min or more indicates that counterregulatory mechanisms would be activated resulting in cortisol release that likely suppresses BDNF expression [209]. The timing of exercise is important to develop optimum change. For an audio–visual example, when executing cognitive tasks just following exercise, the transient increases in BDNF may be utilized and improve memory retention or learning [210]. Additionally, this sort of examination currently has the potential to facilitate the individualized personalization of exercise protocols, which may be especially useful for specific population groups with special needs, such as individuals over 65 years or individuals who inherit the Val66Met polymorphism. Optimized BDNF modulation, and corresponding neurocognitive benefits, are afforded through the use of protected and individualized protocols relative to duration and intensity of activity within these groups. For example, older adults who have undertaken a 12-week HIIT exercise program had 20% increases in hydraulic hippocampal volume and improvements in cognitive flexibility and executive function [211].

In all, findings illustrate the relevance of precision exercise strategies that can be used to promote neuroplasticity and brain health across the life course. Recent studies report that both central and peripheral BDNF signals mediate the neuroplasticity-promoting effects of exercise. Peripheral BDNF is primarily released from skeletal muscle and platelets through exercise and then transported across the BBB via LRP-1 receptors where it influences BDNF central pathways in the hippocampus and motor cortex [212]. Unique molecular-imaging methods evidence the biological relevance of this transporter–transmission mechanism to support synaptic remodeling, neurogenesis, and ’countering’ neurodegenerative processes as a result of exercise. Multiple lines of evidence suggest that BDNF from peripheral tissues meets and integrates with BDNF central signaling, supporting a complex feedback regulatory system that enhances neurotrophic signaling that occurs after exercise (or other forms of physical activity). This highlights exercise as an important ’daily’ decision over the life course for neural health [213].

On a mechanistic level, optimizing mitochondrial function as a result of exercise-induced PGC-1α activation, continues to be one of the central mechanisms of exercise-mediated increases in BDNF. PGC-1α is a regulator of mitochondrial biogenesis, promotes ATP production, and reduces oxidative stress to create an intracellular environment that promotes BDNF synthesis and secretion. Such a process will facilitate synaptic plasticity in normal physiology while reversing the mitochondrial changes in neurodegenerative states. In Alzheimer’s and Parkinson’s disease, for example, aerobic exercise activation of PGC-1α in an ACUTE PHASE decreases challenges with energy deficits and overall neuronal health, while additionally correlating with capacity for increased mitochondrial density, and neuroprotection, stemming from exercise-induced BDNF. Individualized precision exercise programming is already paving the way for optimizing BDNF-modulating approaches. Certainly, HIIT has been shown to be robust for consistently demonstrating the largest increases in circulating BDNF while meta-analysis estimates a 35% increase in BDNF versus moderate-intensity continuous exercise at ~20% [214].

Equally, these effects are also present in people with certain vulnerabilities, such as the Val66Met carriers, which express a pathogenic genetic polymorphism that limits activity-dependent BDNF exocytosis. Thus, aerobic exercise can overcome their hippocampal plasticity limitations and has positively impacted cognitive performance from an exercise intervention for the Val66Met group, to combat their genetic depositions [201]. Structured exercise interventions also provide strong benefits to older adults, with a 12-week exercise intervention demonstrating both increased hippocampal volume and executive function improvement [215]. Overall, with adding in exercise-induced BDNF, including pharmacological and task-specific approaches, creates a highly potent package of neuromodulatory strategies to promote neuroplasticity and neuroplastic recovery across neurological disease contexts. For example, in Alzheimer’s disease, aerobic exercise acts additively to amyloid-targeted therapies to reduce plaque burden and to slow hippocampal atrophy [216].

The combination of treadmill running with aerobic training protects the survival of dopaminergic neurons and enhances the effects of dopaminergic therapies in animal models of Parkinson’s disease, as one stated example because post-stroke rehabilitation is one clear example where this potential for activity enhancers is evident, when task specific intervention is combined with aerobic physical activity. Additionally, serum BDNF concentrations are also vastly increased via this task and aerobic combination approaches, producing re-organization in synaptic plasticity in the primary motor cortex leading to functional motor outcomes [217]. These complementary outcomes expose the translational potential of an exercise + therapeutics approach to recovery and resilience. Furthermore, not only does it have a better understanding of the interplay between intracellular mitochondrial-related mechanism, peripheral BDNF transport, and central neurotropic mechanisms, but it also has the application of exercise-induced BDNF as a neuroprotective strategy. There is potency behind personalized approaches that incorporate individual features, specifically their genomic risk proclivities with age or disease status that are on the verge of transforming lifestyle as an important intervention for treating and preventing neurologic and psychiatric conditions. The corroboration of mechanistic pathways and personalizing exercise-induced BDNF will provide an exercise-mediated intervention to bridge the gap for the prolonging cognitive health and wellbeing and addressing the rising burdens of neurodegenerative diseases.

## 9. BDNF and Diet

### 9.1. Molecular Mechanisms of Nutrient-Induced BDNF Regulation

Dietary bioactive compounds, including polyphenols, omega-3 fatty acids vitamins, and probiotics, regulate BDNF expression primarily through regulation of major molecular pathways, transcription factors and epigenetic mechanisms related to synaptic plasticity and neuroprotection and functioning upon the modulation of intracellular pathways that primarily involve the MAPK/ERK (mitogen-activated protein kinase/extracellular signal-regulated kinase) and PI3K/Akt (phosphoinositide-3-kinase/protein kinase B) signaling pathways mediating the downstream signaling effects of CREB (cAMP response element binding) protein, the most recognized transcription factor in controlling gene transcription of BDNF [218]. Given that many berries, green tea, or cocoa contain polyphenols, which induce significant neuronal protection through the stimulation of TrkB (tropomyosin receptor kinase B receptor) signaling, antioxidant defense, and the modulation of calcium-dependent cellular pathways that facilitate neuronal survival, it is synergistic to note omega-3 fatty acids, specifically DHA (docosahexaenoic acid), tend to increase neuronal membrane fluidity and trigger lipid raft formation necessary for TrkB signaling and clustering. Studies in animal models provide corroborating evidence of a 50% increase in hippocampal BDNF mRNA levels upon DHA supplementation and enhancement in both LTP (long-term potentiation) and cognitive performance as a result [219,220].

Furthermore, vitamins and their effects on BDNF also affect different biochemical and epigenetic pathways through which they modulate BDNF as well. For example, vitamin D induces BDNF release via calcium signaling by upregulating voltage-gated calcium channels, while folate and B vitamins are involved in neuroprotective effects through limiting neurotoxic homocysteine levels, which inhibits neurotrophic signaling [221]. In addition, polyphenolic compounds such as resveratrol and curcumin have an influence on BDNF transcription that is able to bring change to the epigenome by decreasing DNA methylation in the promoter regions thereby enabling overall accessibility to the gene such that the neuroplasticity that it helps to support is maintained [222]. It is important to note that alcohol represents a promising candidate for adverse neurobehavioral and neurochemical neurotoxicity, such that the effects of flavonoids on BDNF expression follows a dose-dependent or hormetic response where moderate doses provided a neuroprotective effect while excessive quantities led to marked oxidative stress, mitochondrial dysfunction, and the induction of intricate inflammatory cascades. A biphasic response such as this also involves nuclear factor erythroid 2-related factor 2 (Nrf2) signaling involved in the regulation of SOD expression, glutathione peroxidase (GPx), and heme oxygenase-1 (HO-1); these help to maintain the redox balance and thus neuronal longevity [223,224,225]. In addition to direct intracellular effects, dietary nutrients interface with the gut–brain axis and microbial metabolism and immune signaling in mediating BDNF expression. With respect to polyphenols, upon being metabolized by the gut microbiome, bioactive metabolites like urolithins (from ellagitannins) and dihydroxyphenyl valeric acid (from catechins), can cross the blood–brain barrier and directly transcribe the BDNF gene in the hippocampus [226].

In fact, increased classical polyphenol intake, preferentially influence beneficial gut microbiota, like Bifidobacterium and Lactobacillus species, which release short-chain fatty acids (SCFAs) that epigenetically increase BDNF expression [227]. This rich interaction between diet, microbial metabolizes, and neurotrophic signaling process reveals the unique role that precision nutrition and identified physiological alterations have in improving cognitive resilience and the ultimate prevention of non-reversible neurodegeneration [228]. Specific polyphenolic compounds, omega-3 fatty acids, and vitamins activate neurotrophic signaling pathways (PI3K/Akt and MAPK/ERK), which result in the phosphorylation of the transcription factor CREB required for expression of the BDNF gene (Figure 2). In addition to providing neuroprotection, these signaling events, promote synaptic remodeling and adaptive responses to offset oxidative stress.

Furthermore, genetic and metabolic variability modifies heterogeneity, and further modifications to the capacity of the diet when BDNF-influencing dietary manipulations should also be considered; the Val66Met polymorphism results in extreme changes in activity-dependent BDNF release due to associable changes in plasticity that can be expressed simply as modifications in neurotrophic effectiveness of the intervention. This may result in an attenuation of neurotrophic signaling within Met-carriers, potentially increasing cognitive dysfunction and psychiatric disorder risk that in turn could benefit from task-specific dietary strategies to compensate for the deficits [229,230].

In addition, diet influences BDNF epigenetic effects such as DNA methylation at BDNF promoter sites. Interestingly, inhibitory BDNF factors due to hypermethylation were recently found to be reversible by consuming hypermethylation-rich diets with methyl donors (i.e., folate, choline, B vitamins) and packing and providing polyphenol food compounds (through histone acetylation) provides a more accessible chromatin state to initiate gene expression [231].

Metabolic function is a critical determinant of diet sensitive BDNF signaling with a potential role for insulin resistance, obesity, and chronic inflammation blocking BDNF-related pathways and critical PI3K/Akt and AMPK signaling to engage synaptic plasticity and neuronal cell survival. The neuroinflammation that disrupts the favorable effects of dietary bioactives is driven by systemic proinflammatory cytokines (e.g., tumor necrosis factor-alpha [TNF-α], interleukin-6 [IL-6]) that inhibit the transcription of BDNF [232].

Hyperinsulinemia and leptin resistance inhibit TrkB receptor stimulation that directly influences BDNF-related synaptic restructuring. The time has come to attempt to improve metabolism by employing glycemic balanced and anti-inflammatory diets to improve the expression of BDNF and restore neuroplasticity to normal limits [233]. At the molecular level, we see that BDNF-promoting diet interventions activate estrogen and the combined actions of the PI3K/Akt and MAPK/ERK pathways that transduce extra-cellular neurotrophic feedforward signals to acquire intracellular transcriptional responses mediating synaptic remodeling and learning and memory consolidation. The best-known downstream effector of PI3K/Akt activation in the neuron is the inhibition of apoptotic death signals and enhancement of CREB-dependent gene transcription, including BDNF transcription, a key molecule for long-term neuronal support. In contrast, the MAPK/ERK pathway is activated by TrkB receptor activation to enhance synaptic connectivity through sequential phosphorylation of Ras, Raf, MEK, and ERK kinases. Phosphorylated ERK then translocates into the nucleus to either directly activate CREB or activate ribosomal S6 kinase (RSK) that up-regulates the expression of BDNF gene [234,235,236].

We can see BDNF as a target for cognitive enhancement and neuro-degeneration prevention, through the wide-ranging neuroprotective and synaptogenic effects of dietary bioactives via these pathways [237,238]. As the exact mechanisms of dietary modulation of BDNF are complex, suggested Macronutrient intake threshold levels (of polyphenols and omega-3s) will require further refinement to produce neuro-protective effects without a metabolic toll. More research into the relationship between gut microbiota in BDNF regulation will also be necessary to provide a framework for dietary strategies that harness microbial metabolism to achieve optimal neurotrophic signaling. Additionally, personalized-balanced dietary-based interventions can also develop from genetic screening, metabolic profiling and epigenetic profiling of an individual’s neuroplasticity requirements, which can yield precision nutrition solutions. Understanding the neurobiological processes that underlie dietary modulation of BDNF will inform approaches to generate effects for cognitive resilience and to ameliorate aspects of neurodegenerative decline [239,240].

### 9.2. Cognitive and Neuroprotective Benefits of BDNF-Boosting Diets

Diets that up-regulate BDNF are co-related with improved cognitive performance in at risk populations such as senescence and in neurodegenerative diseases. Longitudinal cohort studies link dietary patterns that are rich in neurotrophins, with memory recall, cognitive processing speed and reduced likelihood of mild cognitive impairment (MCI) [241,242]. Short-term dietary interventions, such as consumption of flavonoids, are associated with increases in plasma BDNF concentrations in at risk populations that have subsequently been associated with improved working memory and executive function, with benefits reported only a few hours post-ingestion [243].

In pediatric populations, DHA supplementation has had positive effects in terms of cognition and increased attention and academic performance [244]. Moreover, dietary offsets in animal models have revealed increases in synaptic plasticity-related gene expression associated with BDNF up-regulation, improved neural connectivity and adaptive flexibility, and cognitive improvements [245]. In addition to cognition enhancement, it seems there also exists a more universal diet-induced BDNF up-regulation profile which is important for neuroprotection in the context of neurodegenerative and psychiatric conditions. Findings from the clinical scientific literature indicate that chronic consumption of polyphenols reverses pathological signs of AD and does so without experiencing cognitive decline [246,247,248]. In addition, omega-3 supplementation has been found to preserve the functioning of dopaminergic neurons from oxidative stress related to PD and corresponding improvements in motor control [249].

In reference to psychiatric dispositions, patients with BDNF levels were associated with symptom remission of this disposition; with dietary solutions demonstrated to enhance individuals’ mood stabilization and a maximum level of depressive severity [250,251]. Furthermore, noteworthily on the metabolic benefits, ketogenic diets seemed to increase BDNF in the hippocampus, associated with improvements in synaptic stability, and a reduction in acute seizure related to epilepsy [252]. Precision nutrition is an effective way to elevate BDNF and design individualized food-based interventions relative to the inter-individual variability of genetics and metabolic status [208]. Patients with the Val66Met genetic polymorphism (which has been shown to modulate BDNF secretion) respond differently to food and if modulated through dietary means may help simulate restoration of neuroplasticity after injury [253,254]. The development of biomarkers profiling specifically for stoichiometry (regarding homocysteine, and oxidative damage) and ammonia stress would be helpful in creating dietary interventions to maximize BDNF expression retention and neuroprotective outcomes [255].

Intermittent fasting is also a favored intervention for modulating neurotrophic signaling acting as an intermittent stimulus of multi-dimensional structural resilience and functional resistance against neurobiological decline, maybe the most obvious outcomes are related to improvements in memory [256,257]. The commonality is that diverse food-based strategies cousin to a multimodal or synergistic approach in support of cognitive function and neuroprotection and prevention of neurodegeneration. However, a combination consumption of polyphenols, omega-3s, and fasting protocols was found to be more beneficial than single consumption in properties of synaptic plasticity and neuronal longevity. There have recently been significant developments for dietary applications with the ability of the robust and individualized options developed on AI-powered technology with real-time, precision-based recommendations based on a range of genetic and metabolic profiles resulting in applications of precision-based neuroprotective recipes [258]. As such, these integrative multimodal interventions can achieve relevant cognitive and neurological outcomes and be a meaningful and active contributor to the pragmatic, nonpharmacological management of a range of brain-related pathologies and as a safe and low-cost, long-term approach to brain health [259,260].

## 10. BDNF and Stress

### 10.1. Stress Suppression of BDNF: Mechanisms and Consequences

Chronic stress decreases BDNF gene transcription via molecular/cellular processes that impair neurotrophic signaling, compromising brain plasticity. Chronic HPA activation heightens glucocorticoids, that down-regulate phosphorylation of CREB on BDNF promoter IV leading to a regionally specific transcriptional deficit [261]. The hippocampus and PFC demonstrate the most upregulation, with one rodent study demonstrating a 60% decline in hippocampal BDNF mRNA following a prolonged sensitization to stressor(s). Cognitive decline associated with synapse loss, retraction of dendritic arbors, and deficits in learning/memory underscores the importance of BDNF signaling in driving and maintaining neural networks [262].

Glucocorticoids also extinguish these responses by inducing oxidative stress, mitochondrial injury, and depleting local energy stores critical to synaptic plasticity. Additionally, under stress microglia activate and release proinflammatory cytokines (i.e., IL-1β, TNF-α), which further inhibit BDNF signaling [263]. In one study, authors discovered that a microglial activation blocker ameliorated the decline in hippocampal BDNF and restored spatial memory deficits driven by stress. These studies highlight the multi-factorial processes that underlie BDNF gene expression suppression by stress and ultimately lead to deficits in global neuroplasticity and increased risk for neurocognitive/emotional dysfunction [264].

### 10.2. BDNF’s Role in Resilience to Stress

Not only does baseline BDNF play a role in the outcome of stress, but higher BDNF has the protective effect of preserving synaptic integrity and counteracting the response to chronic stress. Facilitating TrkB receptor activity in the PFC and hippocampus enhances synaptic plasticity and facilitates constructive neural remodeling and preserves ecological niche [265]. In humans, the studies related to serum BDNF levels and HPA axis reactivity to acute stress over time show a negative association which suggests BDNF may modulate HPA axis hyper-responsivity [266]. In animal studies using preclinical models, there is further indication of a protective role for BDNF. For instance, viral-mediated overexpression of BDNF in the PFC protects from stress-induced synaptic deficiencies by sustaining dendritic spine density and cognitive flexibility [267].

Polymorphisms like Val66Met, highlight BDNF’s role in resilience to stress. There are cases where such an imbalance has led directly to psychopathology; parallel data in mood and anxiety disorders had there been the identification of BDNF as a candidate gene in mood disorders in individuals carrying the met allele, who have a reduced activity-dependent BDNF release and, therefore, increased susceptibility to stress-induced mood disorders, and there would be indicated a possible mechanism in which signaling compensatory upregulation may prove advantageous [268].

### 10.3. Stress-Related Psychiatric Disorders: BDNF Dysregulation as a Mechanistic Link

The dysregulation, or aberrant regulation, of BDNF has also been identified as a vital mechanism of pathophysiology for stress-related psychiatric disorders, e.g., MDD, generalized anxiety disorder (GAD), post-traumatic stress disorder (PTSD), etc. [168]. The chronic stress-mediated suppression of hippocampal BDNF levels suppresses synaptic plasticity and links to cognitive dysfunction and emotional dysregulation in MDD. Clinically, patients with MDD show serum BDNF levels that are associated with the most significant declines in treatment-resistant patients [269]. Various antidepressant treatments (including SSRIs) indirectly increase BDNF, demonstrating stronger increases in plasma BDNF levels with relief of symptoms throughout treatment [270]. In PTSD, stress induced repression of the amygdala BDNF message disrupts neuronal circuits for fear extinction and emotional regulation. In preclinical models, normalization of fear extinction, and decreases in hyper vigilance were observed with viral mediated over-expression of BDNF in the basolateral amygdala [271]. In the same way, the decreased secretion of BDNF in the PFC would negate the regulatory influence of the PFC on the amygdala and enhance emotional reactivity and stress sensitivity found in GAD. Collectively these findings implicate BDNF as a mechanistic link between chronic stress and the development of both physical and mental illness and warrants investigation as a potential target for therapy [272].

### 10.4. Interventions to Restore BDNF and Mitigate Stress Effects

One intervention aimed at improving neuroplasticity and resilience would be to rescue the BDNF expression that is reduced by chronic stress. Aerobic exercise is one of the most powerful pharmacologically based first line interventions, while moderate-high intensity physical activity can elevate serum BDNF by 20–30% along with up-regulating indicators of synaptic density in the hippocampus [273]. In individuals, exercise raised BDNF and reduced cortisol, resulting in significant improvement of executive functioning and emotional regulation in a 12-week clinical trial with highly stressed individuals [274].

All these pharmacological strategies directed at BDNF signaling are receiving increased consideration. To name a few, LM22A-4, a selective TrkB agonist that directly engages BDNF pathways to support synaptic repair and recover from cognition dysfunction due to stress, in preclinical models [275]. Rapidly acting anti-depressants include ketamine, which forms a part of our next trolling club, by increasing BDNF, via mTOR mediated protein synthesis, to induce synaptic re-modeling in hours. Transcranial magnetic stimulation (TMS) has also shown an increase in plasma BDNF with a promise of a non-invasive therapeutic approach for stress-related conditions [276]. Emerging approaches include mindfulness-based stress reduction (MBSR), specifically an 8-week MBSR, which resulted in a 20% increase in plasma levels of BDNF, coupled with an improvement in strategy-mediated stress reactivity, and improvement in emotional regulation. These multimodal approaches, integrating behavioral, pharmacological and lifestyle interventions, provide a comprehensive system of normalizing BDNF, to mitigate the impact of stressors to develop resilience [277].

## 11. BDNF and Pharmacological Interventions

### 11.1. TrkB Agonists: Directly Enhancing Neurotrophic Signaling

If your goal is to directly and specifically augment deficits caused by BDNF, TrkB agonists present the most appropriate means of activating TrkB, the receptor that mediates neurotrophic signaling and ultimately leads to synaptic plasticity. Essentially, TrkB agonists activate TrkB directly and initiate physiological processes consistent with BDNF, mostly by activating the same major neurotrophic signaling pathways for neuronal survival and neurogenesis (i.e., PI3K/Akt; MAPK/ERK; PLCγ) [278]. In particular, LM22A4, the small molecule TrkB agonist with neuroprotective efficacy in preclinical models of neurodegeneration, also induced AMPA receptor trafficking alongside promoting increased synaptic incorporation of AMPA receptor through the increase in AMPA receptor trafficking, which also improved spatial memory performance in models of AD treated at the same time as LM22A4 [279].

Cyclic peptides, with a nextgen understanding of limitations in stability of receptor-binding specificity in small molecules, have also induced robust enhancement of axonal transport, and decreased oxidative injury in preclinical models of ALS and increased duration of motor neuron viability [280]. With regard to PD specifically, cyclic peptides also restored mitochondrial function and decreased apoptotic neuronal viability of dopaminergic neurons while also influencing motor ability in other models of PD. The potential of being able to formulate new cyclic peptide-based treatments to cross the BBB also showed that TrkB agonists possess therapeutic versatility and potential [281].

### 11.2. BDNF Modulation by Antidepressants: Broadening Mechanistic Understanding

Interestingly, despite the fact that BDNF is impacted by a variety of mechanisms other than through monoaminergic signaling, antidepressants constitute one of the cornerstones of indirect therapeutic mechanisms to regulate BDNF expression with the bulk of the treatment effects derived from the interactions with monoaminergic signaling in its broadest sense. The selective serotonin reuptake inhibitors (SSRIs) and selective norepinephrine reuptake inhibitors (SNRIs) elevate levels of 5-HT and NE in the extracellular space and activate intracellular signaling pathways, which universally lead to CREB phosphorylation and BDNF transcription from its gene [282]. Clinical studies have shown that there is consistently around a 25–30% increase in plasma BDNF levels in individuals responding to SSRIs/SNRIs treatments, and this increase in plasma BDNF levels is correlated with symptom remission of depression. Structural MRI studies have demonstrated that BDNF-induced increases after antidepressant treatment will spare any structural consequences of depression, specifically the reduced structural volumes of the hippocampus [283]. The mentioned depressive treatment interventions (i.e., intravenous and rapid-acting antidepressants), specifically ketamine, are one factor that has introduced a shift in the treatment paradigm of depression. Other than the acute ketamine treatment allowing restoration of a measurable capacity of neuroplasticity in the PFC, which is capable in hours, it is related to NMDA receptor antagonism, and the excess activation of mTOR resulted in rapid synaptogenesis (with an increase in PFC connectivity). GLYX-13 (rapastinel) has similar effects as ketamine, acting as a partial NMDA receptor agonist, albeit with no dissociative features as with ketamine as an established new pharmacological intervention for treatment-resistant depression. The presence of these new interventions reinforces the role of BDNF for mood disorders, especially in those patients who do not respond to treatment in the traditional manner [284,285].

### 11.3. Neurodegenerative Applications: Addressing Synaptic and Cellular Loss

As a pharmacological drug, LDL intervenes with the BDNF mechanism in the body; now in conjunction with these new pharmacological interventions, a new paradigm for treating neurodegenerative disease is warranted. For AD mentioned above, one of the effects of BDNF is the stability of the synapse via AMPA and NMDA receptor trafficking and disallowing tau hyper-phosphorylation [286]. The gene delivery of BDNF with AAV has extraordinary potential in pre-clinical models in reducing amyloid-β deposition and increasing with the preservation of hippocampal connectivity. Imaging studies with loss-of-function mutations of BDNF have shown that BDNF treatments related to patterns of disorganized/deficient hippocampal activity suppress hippocampal disconnections, which improved episodic and spatial memory [287].

For PD, the activated TrkB neuroprotection of dopaminergic neurons occurs via a decrease in oxidative stress, more stability in mitochondrial stability, and preventing apoptosis. These studies have also demonstrated a synergism of TrkB agonists with dopamine replacement therapies, with a 40% enhancement of motor function compared to monotherapy [288]. BDNF conjugated pharmacological agents that increase BDNF transcription/transport stability have worked to provide protection to corticostriatal circuits and restore motor function in HD, despite BDNF transcriptional/transport deficits. Thus, BDNF-targeted therapies are among the cornerstones for neurodegeneration [289].

### 11.4. Emerging Therapies: mRNA Delivery and Gene Editing

Novel methodologies like mRNA-based therapeutics and gene editing are changing the field of modulating BDNF. One added advantage of utilizing lipid nanoparticle-based platforms is the ability to administer BDNF mRNA directly to a targeted tissue without crossing the BBB, resulting in functional BDNF production in situ where contextual need arises [290]. Specifically, in AD models, treatment with BDNF mRNA led to a more than 200% increase in hippocampal BDNF and reversed synaptic deficits, significantly improving memory within weeks. Collectively, these systems resolve fundamental delivery issues and allow for a modular treatment for the global neurotrophic deficit problem [30].

Gene editing technologies, such as CRISPR-Cas9, are enhancing BDNF modulation for precision and specificity, by utilizing targeted endogenous promoters for up regulation of transcription. For example, in PD models CRISPR activation of BDNF by CRISPR protected dopaminergic neurons and motor function was preserved [291]. Epigenetic editing technologies (e.g., dCas9 fusion proteins) sustain longer duration BDNF expression changes by regulating chromatin access that generates persistent neurotrophic effects without further treatment intervention. These technologies represent a paradigm shift for neuroprotection and provide the opportunity for longer-term personalized therapies for complex neurological diseases [292,293].

Emerging therapeutics with rapid acting effects such as ketamine and rapastinel are transforming the therapeutic landscape for mood and neurodegenerative disorders. These interventions support synaptic remodeling and facilitate more synaptic connectivity, in an order of hours, by rapidly restoring BDNF levels and integrating downstream mTOR pathways. These agents have unmatched potency for treatment-resistant depression, and both ketamine and BDNF restore prefrontal cortical BDNF to the same range, and stress induces similar over coupling of deranged neural network [294]. Some of the best indications have been emerging combinatorial strategies that are synergizing BDNF-targeted strategies with the standard of care. TrkB agonists, in addition to BDNF stabilizing agents, and other agent classes are insistingly reported to prospectively augment therapeutic outcomes of dopamine replacement in PD, and AAV delivery of BDNF to potentially augment amyloid-clearing mechanisms in the treatment of Alzheimer’s disease. These synergies not only enhance therapeutic outcomes but also expand the number of BDNF-targeted therapeutics for solely multi-causative six neurological syndromes [295].

## 12. BDNF as a Therapeutic Target for Precision Medicine

Controlling neuroplasticity seems to be a more important target for precision medicine; BDNF plays a large role in neuronal survival, synaptic remodeling, and cognitive ability. The BDNF signaling has shown relevance to neurodegenerative diseases and psychiatric disorders where solutions may be tailored to meet genetic and molecular specificities across individuals. Evidence indicates from human and animal data that the Val66Met polymorphism, which alters the intracellular trafficking and activity pedigree of BDNF, is a primary variant that signals status of BDNF availability. Individual cognitive outcomes associated with this variant have been observed with deficits of hippocampal plasticity, as well as concomitant with greater risks for general cognitive impairment and affective disorder [296,297].

In the presence of such heterogeneity, TrkB modified therapeutic agents, as well as the use of epigenetic modulator interventions may be part of the solution to restore neuroplasticity and slow disease progress [298]. Current genomics and epigenetics profiling permits clustering of patients by individual variations in single nucleotide polymorphisms (SNPs) that recognize genetic specifications around the trafficking, expression, and downstream signal response of BDNF [299]. Also, epigenetic dysregulations such as hypermethylation in BDNF promoter regions support the relevant exploitation of targeted precision therapies for restructuring gene expression in complicated or complex scenarios around mood and neurodegeneration [300].

Thus, proteomics also extends these works in linking the emergent synaptic to mitochondrial pathology nets into the BDNF level, conferring specificity and meaning to a continuously trending (progressive) functional biomarker that identifies a hitherto commonly protective phenotype that defines neurons’ resiliency (or independence) to death. The combination of BDNF with tau and amyloid proteins in some studies has greatly heightened the accuracy of very early diagnostic approaches in respect to Alzheimer’s disease [301]. New biosensor technologies allow for real-time assessment of BDNF, having the potential to resolve the variability inherent in dynamic BDNF fluctuations that clinicians will require to remediate and personalize their therapeutic devices [302].

This advancement and proliferation in therapeutic areas that target the manipulation of BDNF has created opportunities to significantly reposition the management options for neuroplasticity-based therapies utilizing exciting new frameworks of precision pharmacotherapy, gene-editing strategies and machine learning/AI-based diagnostic approaches. Other pharmacological avenues for increasing BDNF signaling include TrkB agonists, such as cyclic peptides and small molecule drugs, to provide direct stimulation of TrkB receptors that are independent of the many BDNF secretion deficits that exist in many genetic models. Cyclic TrkB agonists produce significant motor function repair and protection of dopaminergic neurons from injury such as those that occur in PD in preclinical models and which compare favorably to standard of care dopamine replacement medications [112,303]. Gene-editing technology, particularly CRISPR-Cas9, and other related genome editing technologies can now be used to implement the precise induction of BDNF gene transcription by altering transcriptional regulation elements for the BDNF growth factor. Applications of CRISPR-mediated induction of BDNF have been found to successfully restore corticostriatal connectivity in models of Huntington’s disease, and there is even evidence that they result in decreased neuronal loss and increased life-span [304]. Conversely, lipid nanoparticle-delivered mRNA-based therapeutics have shown an ability to remediate synaptic deficiencies and memory impairments that are present in AD, thus providing a scalable method for improving neuroprotection [305].

The incorporation of dynamic biomarker monitoring to adjust interventions is another factor that is increasing the flexibility of BDNF-type interventions. Use of wearable biosensors that have the ability to measuring plasma BDNF, cortisol, and indicators of cognitive function, provide the ability to make adjustments to therapy on more useful time scales to improve efficacy through monitoring feedback [306]. At the same time AI-based precision medicine platforms are deploying individualized patient genetic, epigenetic, and proteomic data to predict individual likelihood of responsiveness to TrkB agonists, gene therapies, or pharmacological drugs to inform treatment options and the monitoring of prognosis [307,308]. Future studies will emphasize combinatorial therapies targeting multiple disrupted pathways simultaneously. Aspects of these dual-action delivery systems with a co-administration of CRISPR constructs and mRNA therapies concurrently are in development to support the long-term restoration of BDNF while with the risk mitigation of TrkB agonists with mitochondrial enhancers and anti-inflammatory agents to potentially target more global neurodegenerative mechanistic backgrounds [309]. In addition, patient stratification is taking further shape with active research groups developing more reliable epigenetic mapping and multiplex biomarker panel approaches to substantiate biological endophenotypes such as BDNF hypermethylation that serve as biological hieroglyphics for cognitive decline or mood disorders. The emerging field of introducing BDNF driven precision medicine from adaptive integrated interventions that synthesize gene therapy, neurotrophic pharmacology, and AI machine learning-based treatment optimization is beginning [12,310].

As all aspects of real-time biomarker measurements combined with AI machine-learning individualized treatment phases can be demonstrated, this will begin the era of second generation BDNF strategy targeting therapies that result in simultaneously and completely changing the treatment trajectories for neurodegenerative and psychiatric illnesses [311,312,313,314].

## 13. Challenges and Future Directions in BDNF Research and Therapeutics

While BDNF presents a potential for therapeutic benefits for a variety of neurological disorders, clinical application of BDNF remains limited due to delivery difficulties, patient-to-patient variability, and complicated measurement. BDNF has a chemical structure which limits the efficiency of crossing the BBB and must be delivered via an invasive route, which precludes large-scale clinical use. Additionally, delivery by systemic administration has low bioavailability and high enzymatic degradation suggesting the use of nanoparticle carriers, viral vectored systems, and gene-editing methods to enhance endogenous BDNF production [315,316]. In addition to delivery, patient-to-patient variability of what is performed and the activation of BDNF during brain plasticity will reduce the result of the therapeutic success. Genetic polymorphisms such as Val66Met can change neurotrophic signaling, which can negate synaptic plasticity and treatment outcome. Similarly, epigenetic modifications, such as promoter hypermethylation will inhibit BDNF transcription and changes your ability to achieve symptom remission or illness control. The variability between polytypes and epigenetics must utilize precision-medicine to customize drug therapy for individual patients within defined molecular profiles [317].

Additionally, too much stimulation of the TrkB receptor can also cause maladaptive plasticity and excitotoxicity which all demonstrate the need for spatial-temporally defined, brain-regional-specific modulation of TrkB receptor for it to have maximum beneficial efficacy, thus, demonstrating the difficulty in precisely measuring potential BDNF modulated states and disorders [318]. Overall, the lack of precise measurements is a major weakness in the BDNF literature. Assay analysis using a standard measurement cannot tell us with specificity whether BDNF levels are indicative of mBDNF (modulating neuroplasticity) or proBDNF (that is promoting synaptic pruning leading to identifying mBDNF and proBDNF as brain-based features for categorizing disorders in neuroscience and psychiatry) [319]. While advanced detection technologies (e.g., digital ELISA, mass spectrometry) are more specific, they are still limited in application due to the limitations of high cost and technical complexity [320]. In addition, peripheral BDNF levels, the most commonly used biomarker, cannot be relied on to accurately represent CNS activity due to systemic factors (e.g., inflammation, metabolic alterations, and platelet activation), thus stressing the need for direct neurotrophic biomarkers [321]. On top of this are the translational limitations of pre-clinical models. Using a rodent model system can generate variability in genetic make-up, and in neural network organization; however, genetic differences and network differences are the main sources of variability that are providing low predictability of success of BDNF for clinical models. The use of human transcription profiles in both iPSC-derived neurons and brain organoids has created a more relevant model for BDNF regulation, and for defining the vectors of treatment response in the human brain. Nevertheless, they are limited to special applications due to scaling, reproducibility, and cost considerations [120,322].

In terms of clinical utility, there would have to be concern over barriers of delivery, gene regulation and precision diagnostics to map a BDNF-based therapy. Examples include NP delivery systems that utilize LNPs that encapsulate BDNF mRNA to promote local synthesis of protein and that evade enzymatic degradation and the pressures exerted by the BBB [323]. Second-generation AAV vectors have been altered and provided with unique tissue-specific promoters to reduce immunogenicity and will continue to provide long-lasting BDNF expression, subsequently aiding in neuronal connectivity and promoting survival across multiple models of neurodegenerative disease [324]. Advances in gene editing, as well as advances in epigenetic modulation allow for activity-specific control of BDNF. Neuronal protection, as well as functional recovery, can be induced in PD by targeting certain regulatory elements using CRISPR-Cas9, as well dCas9-mediated histone acetylation is an example of a robust epigenetic editing tool capable of etching stable activation of transcriptional priority with a regulative capacity capable of restoring BDNF deficits common in mood and stress-related disorders [325]. Moreover, it seems that these effects are augmented with the addition of histone deacetylase (HDAC) inhibitors which improved the expression of BDNF and neuroplasticity deficits, grounded in preclinical studies [326]. Mechanisms representing this contribution are presented in Figure 3, supporting transcriptional dysregulation, nuclear accumulations, and calcium homeostasis as inter-related components of the neurodegenerative process. The clinical significance of HDAC inhibitors and BDNF mimetics with being BDNF therapy targets are that they disrupt these pathological cascades and promote restoration of neuronal function as illustrated in the figure.

Notably, multiplex biomarker platforms that incorporate BDNF with neurodegenerative markers improve clinical diagnostic accuracy in early disease screening and treatment follow up [327]. Therefore, the development of biosensors that can separate mBDNF from its pro-form allows for a better assessment of neurotrophic activity, and ultimately drug efficacy. Humanized preclinical models of BDNF regulation and genetic variability, such as iPSC-derived neurons and brain organoids support the evaluation of BDNF regulation in a patient-specific manner, aligned with efforts developing personalized interventions [328]. Emerging AI-enabled platforms are further transforming the precision medicine landscape for BDNF-directed therapies by integrating genetic, epigenetic, and biomarker data to determine responses to treatment and specify therapeutic approaches for each significant population complement [329]. As such, biosensors or wearables that can continuously assess plasma BDNF levels and stress biomarker levels, along with cognitive performance will emerge as an innovative method to identify opportunity for real-time adjustments to the therapeutic intervention that may maintain therapeutic efficacy [330].

The adoption of these assessment platforms with telemedicine offers new and unique opportunities to achieve BDNF-targeted therapies across different clinical contexts outside of hospital settings. The next frontier in therapy innovation, is in combinatorial therapies directed at more than one neurodegenerative pathway. Dual-action systems, such as nanoparticle-delivered therapy capable of simultaneously delivering CRISPR constructs and BDNF mRNA will restore gene expression and protein synthesis together. Targeting TrkB agonists with mitochondrial enhancers and anti-inflammation will provide increased neuroprotection by targeting both upstream and downstream neuro-pathogenic processes [154].

The vocabulary and discourses surrounding integrated synergistic treatment approaches currently constructs the next generation of neurotherapeutics advancing BDNF as a central underpinning axis across neurodegenerative and psychiatric disorders.

## 14. Conclusions and Future Directions—The Roadmap for BDNF-Centered Therapies

BDNF has become much more than a neurotrophic factor: it has emerged as an intellectual lynchpin of clinically relevant modern neuroscience bridging molecular biology/precision medicine with systems-based therapeutics. Given its remarkable ability to modulate synaptic plasticity, enhance cognitive resilience, and reinvigorate neuroplasticity, BDNF represents a superordinate regulator of brain health while simultaneously emerging as an unwitting accomplice in the pursuit of resolving some of the more difficult puzzles regarding human behavior in neurology and psychiatry. These new BDNF-based therapies threaten to disrupt the traditional story around the disease pathophysiology that moves from synaptic failure in AD to poor stress tolerance in MDD; they may also result in more disease-modifying options that allow for an approach to treatment beyond symptoms. The next decade is poised to deliver innovations that might have seemed impossible just a few years ago. New delivery systems are poised to provide safe, sustained, and stable levels of BDNF in the human brain, delivered via lipid nanoparticle-mediated mRNA therapies or CRISPR-Cas9-mediated epigenetic editing. These innovations promise to assist in overcoming long-standing challenges relating to bioavailability, off-target effects, and blood–brain barrier penetration in order to make real-world BDNF-directed therapies a clinical reality. Moreover, diagnostics and therapeutics will be further integrated within artificial intelligence-enabled platforms and wearable biometric sensors to provide a whole new layer of care—real-time adaptive care systems. This facilitator will spark a new era of precision medicine where BDNF levels will provide an ongoing measure of treatment response, interventions can be individually titrated on the fly, and patients will have much more active roles in the overall regimen of treatment. As we widen our nets, BDNF will have a much more expanded footprint, likewise, outside of the CNS. Emerging knowledge of its effects on the gut–brain axis, systemic inflammation, and cardiovascular health paves the way for an innovation in functional healthcare. Treating disease using the BDNF boosters is not only the potential of BDNF interventions, but even further, it could enhance resilience, cognitive longevity, and wellbeing throughout life. Whether it be neurodevelopmental health promotion in children or treatment for age-associated cognitive decline, the possibilities of BDNF are limitless. However, the promise of this future involves more than only scientific advancements; it involves ethical accountability, global equity, and interprofessional collaboration. Working with advanced tools such as CRISPR-dCas9, mRNA therapies and real-time monitoring systems means there should be interventions and industrial governance tied into all that we do to ensure safety, transparency, and trust in the process. Equally important is that these technological advancements, including therapies targeting BDNF levels, are warranted for every person, from every place and all socioeconomic circumstances.

All these discoveries could be scaled up collaboratively between government, academia, and industry so that BDNF-centered therapies are not just a gift of the few but rather a gift of the many—all working together. This is an arena where the stakes are high; BDNF is one of the few foundational molecules that is turning itself into an integrative axis of preclinical treatment, not just of neurological and psychic disorders that can include psychosis and schizophrenia but also the pre-diseases and pre-conditions of rethinking that can lead to them, including childhood. Achieving a convergence of innovation, equity, and ethical rigor in the coming years will be needed, but the potential is undeniable: BDNF-based therapies could alter the course of brain health for many years to come. We are not only revolutionizing the treatment paradigm with BDNF—we are in fact constructing medicine’s future. Challenges to clinical deployment of BDNF-directed therapies However, there are still challenges to the clinical deployment of BDNF-directed therapies. Most importantly, BDNF signaling is tightly regulated, and both hypo- and hyper-signaling may have pathological ramifications. Moderate neurotrophic signals may induce neuroplasticity; however, on the contrary, overstimulation (ranging from TrkB agonistic compounds) could lead to excitotoxicity or impair neuronal homeostasis. Success of these treatments will entail establishing ideal therapeutic windows and optimizing dosing regimens for efficiency with minimal side effects. Like all BDNF-based therapies, BDNF-directed treatments have a non-negligible opacity hurdle as they cannot penetrate the BBB without assistance.

Even the most promising methods relying upon recombinant BDNF proteins or gene-acquisition strategies all have low bioavailability and off-target effects. Questions regarding long-term safety, immunogenicity, and scalability have yet to be addressed in new delivery methods, including lipid nanoparticles and viral vectors in the future clinical pipeline. Interindividual variability is another parsimonious challenge. Genetic considerations such as Val66Met polymorphism that influence BDNF release and synaptic modulation introduce common treatment response heterogeneity. They also build from epigenetic shifts that could evade in silico prediction of treatment responses and possible metabolic constructs that could become the basis for personalized treatment paradigms for BDNF-directed therapies. The use of peripheral BDNF as a CNS functional biomarker warrants ongoing debate. Because of the influence of peripheral factors, including systemic inflammation and platelets, serum and plasma levels are not centrally specific. More definitive approaches will include analyzing cortex-derived BDNF via cerebrospinal fluid and neuroimaging-based analyses of the level of BDNF influence from therapy. Although ethical considerations with regard to safety and accessibility, as well as data privacy with precision medicine tools ranging from CRISPR to AI-driven diagnostics to real-time monitoring, cannot be consigned to the back burner. Without equitable access and regulatory access to these innovations, we risk their use becoming exclusively available through payment, making them only available to a select and elite cohort of the population, thus compounding health discrepancies.

To ensure the challenges can be addressed, allowing for clinical translation of BDNF-targeted therapy is critical. With technical advances in delivery strategies, specificity of intended targets of application and evidence-based inquiry in their conclusions of use, these paradigm-altering advances can move from intellectual endeavors to clinical-scale products.

## Figures and Tables

**Figure 1 ijms-26-04271-f001:**
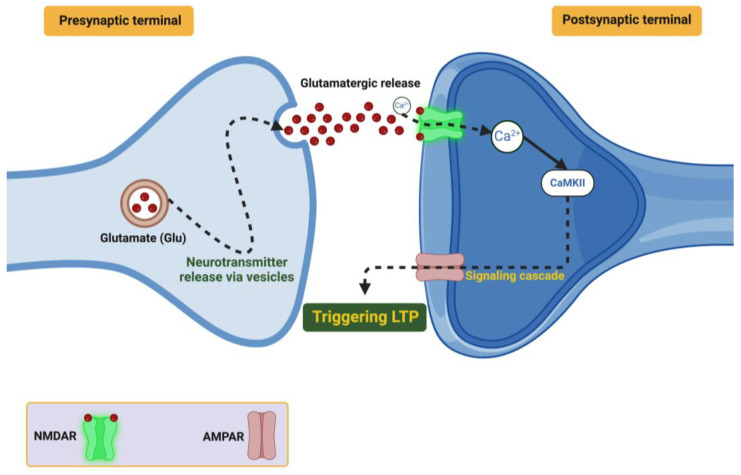
Demonstrates how glutamate (Glu), released from presynaptic vesicles, binds to NMDA and AMPA receptors on the postsynaptic terminal. NMDA receptor activation, facilitated by calcium (Ca^2+^) influx, triggers downstream signaling pathways involving CaMKII and other kinases, which lead to LTP.

**Figure 2 ijms-26-04271-f002:**
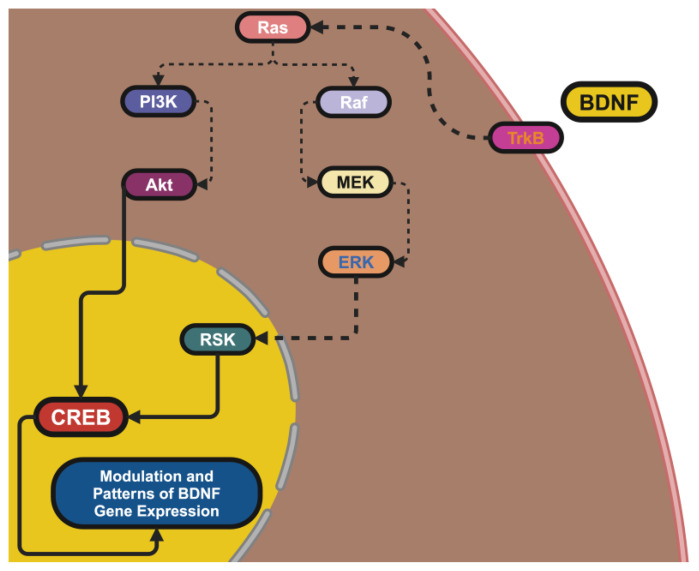
Provides a schematic representation of the molecular pathways through which dietary bioactives regulate BDNF expression. Upon activation by BDNF, the TrkB receptor initiates two primary signaling cascades that play a pivotal role in neuronal function and survival. The PI3K/Akt pathway is a critical mediator of neuroprotection and synaptic plasticity. The activation of PI3K leads to the phosphorylation of Akt, a kinase that enhances neuronal survival by inhibiting apoptotic pathways and promoting cell growth. Akt also plays an indirect role in BDNF regulation by activating downstream transcription factors, including CREB, which binds to BDNF promoter regions and enhances its transcription. This pathway is particularly sensitive to dietary modulation, as polyphenols and omega-3 fatty acids have been shown to amplify PI3K/Akt signaling, thereby increasing BDNF levels and supporting neuroplasticity. The MAPK/ERK pathway serves as another key regulatory mechanism for BDNF expression. Upon TrkB activation, the small GTPase Ras initiates a signaling cascade that sequentially activates Raf, MEK, and ERK kinases. The phosphorylated ERK translocates into the nucleus, where it facilitates CREB activation either directly or through the ribosomal S6 kinase (RSK). This process enhances synaptic remodeling, learning, and memory formation, making it a crucial target for dietary interventions that aim to bolster cognitive function. Together, these pathways highlight the intricate molecular interplay between diet and neurotrophic signaling. By targeting these mechanisms, dietary interventions can serve as powerful tools for enhancing BDNF activity, supporting brain health, and mitigating the effects of neurodegenerative diseases.

**Figure 3 ijms-26-04271-f003:**
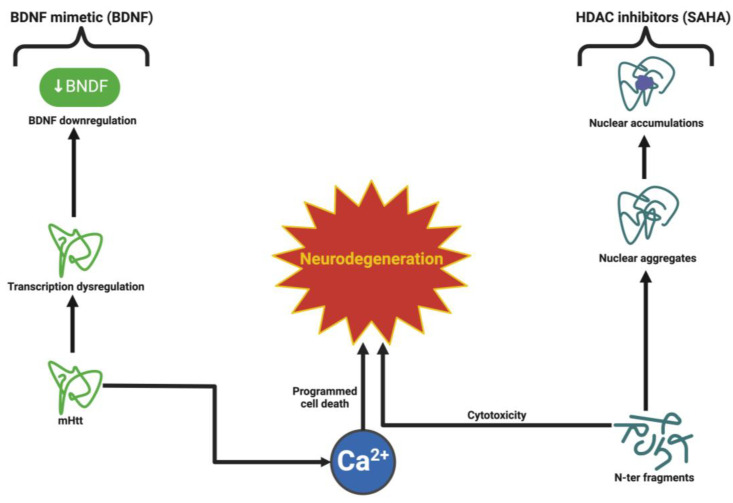
Epigenetic modulation and molecular mechanisms in neurodegeneration. This figure provides an overview of the molecular pathways contributing to neurodegeneration, with a focus on BDNF dysregulation and its downstream effects.

**Table 1 ijms-26-04271-t001:** Provides a comprehensive overview of BDNF’s emerging roles in diagnostics, molecular mechanisms, and systemic functions beyond its traditional focus in the central nervous system.

Research Area	Study Goal	Methodology	Major Findings	Impact and Implications	Citation
BDNF as an Early Alzheimer’s Biomarker	Evaluate plasma vs. CSF BDNF levels for early detection.	Longitudinal human study with biomarker tracking.	Plasma BDNF declines 10 years before clinical AD onset, correlating with hippocampal atrophy and CSF levels (R^2^ = 0.82).	Supports plasma BDNF as a non-invasive predictor, improving early diagnosis.	[21]
Exercise-Induced BDNF Transport Across the BBB	Determine how peripheral BDNF reaches the brain post-exercise.	Rodent treadmill study with BBB permeability assays.	LRP-1 receptor mediates BDNF transport, enhancing hippocampal plasticity and memory by 40%.	Establishes exercise as a mechanism to enhance central BDNF levels.	[22]
Stress, Epigenetics, and BDNF Regulation	Investigate the effects of chronic stress on BDNF gene expression.	Human cohort with DNA methylation analysis.	Hypermethylation of BDNF promoter IV correlates with hippocampal shrinkage; demethylation reverses resilience deficits.	Highlights epigenetic interventions as potential psychiatric treatments.	[23]
Gut–Brain Axis and BDNF in Inflammation	Assess how gut inflammation affects neuroplasticity.	Germ-free mouse model with induced colitis.	Colonic inflammation reduces systemic BDNF, impairing neurogenesis and cognition; probiotics restore function.	Suggests gut microbiota modulation as a neuroprotective strategy.	[24]
BDNF and Cardiomyocyte Survival Under Stress	Explore BDNF’s role in cardiac adaptation to hypoxia.	Cardiomyocyte cultures and ischemic rodent models.	Hypoxia triggers BDNF upregulation, promoting mitochondrial protection and cell survival via TrkB-PI3K activation.	Expands BDNF’s role beyond the CNS, identifying cardioprotective mechanisms.	[25]
BDNF in Vascular Repair During Systemic Inflammation	Examine its function in endothelial regeneration.	Preclinical sepsis model with vascular integrity assays.	BDNF enhances VEGF expression, improving endothelial stability and reducing inflammation-induced leakage.	Positions BDNF as a vascular repair mediator in inflammatory conditions.	[26]
Sex Differences in BDNF and Neuroprotection	Identify gender-based variations in BDNF dynamics.	Human cohort.	Women maintain higher BDNF levels, linked to lower neurodegeneration risk; estrogen enhances TrkB signaling.	Emphasizes sex-specific approaches in neurodegenerative disease prevention.	[27]
Differential Roles of proBDNF and mBDNF in Neurodegeneration	Analyze distinct effects of BDNF isoforms on disease progression.	Mass spectrometry of CSF samples in AD patients.	ProBDNF elevation signals early neurodegeneration, driving synaptic loss and inflammation.	Establishes proBDNF as a biomarker for early-stage neurodegenerative decline.	[28]
BDNF and Sensory Neuron Function in Diabetes	Assess BDNF’s role in diabetic neuropathy.	Patient cohort study with sensory function assessments.	Low systemic BDNF correlates with increased sensory deficits and impaired pain response.	Highlights BDNF’s therapeutic potential for neuropathy management.	[29]
BDNF and Tumor-Associated Neural Plasticity	Investigate its role in neural adaptations in cancer.	Tumor-derived nerve culture models and clinical biopsy analysis.	Cancer-secreted factors upregulate BDNF, promoting tumor-related axonal sprouting.	Identifies BDNF as a potential therapeutic target in neuro-oncology.	[30]

**Table 2 ijms-26-04271-t002:** Summarizes cutting-edge studies on BDNF-related mechanisms and therapies. It includes a diverse range of experimental approaches, from preclinical animal models exploring mechanisms of BDNF regulation to clinical trials testing novel therapeutic strategies. The table highlights translational breakthroughs, such as nanoparticle-based BDNF delivery, optogenetic activation of BDNF signaling, and dietary interventions, showcasing their relevance to modern precision medicine.

Therapeutic Approach	Target Condition	Key Outcome	Novel Contributions	Reference
LNP Delivery of BDNF mRNA	Alzheimer’s Disease	↑ Synaptic density (200%), reversal of memory deficits in 4 weeks	First non-invasive BDNF mRNA therapy, overcoming BBB and degradation issues	[131]
CRISPR-Based BDNF Gene Activation	Parkinson’s Disease	70% dopaminergic neuron survival, 50% motor improvement	First CRISPR-dCas9 approach for long-term neuroprotection without direct BDNF protein delivery	[132]
Aerobic Exercise and Serum BDNF	Cognitive Aging	↑ Serum BDNF (30%), ↑ Executive function (35%)	Direct evidence of exercise-driven BDNF elevation improving cognitive resilience	[133]
Epigenetic Modulation via HDAC Inhibitors	Depression	↑ Hippocampal BDNF (60%), reversal of behavioral deficits	Established epigenetic repression of BDNF as a drug target for depression	[134]
Cyclic TrkB Agonists	ALS	↓ Motor neuron loss (50%), ↑ Motor function (40%), ↑ Survival (25%)	Developed BBB-penetrant TrkB agonists for motor neuron preservation	[135]
Plasma BDNF as a Biomarker	Preclinical AD.	BDNF decline 10 years pre-symptom onset, correlated with hippocampal atrophy (R^2^ = 0.82)	Identified plasma BDNF as a non-invasive early biomarker for AD	[136]
Viral BDNF Delivery in the Amygdala	PTSD	↑ Fear extinction (45%), ↓ Hypervigilance	Demonstrated region-specific BDNF upregulation for fear extinction therapy	[137]
BDNF mRNA Therapy	Huntington’s Disease	↑ Spine density (50%), ↑ Motor function	First mRNA-based BDNF restoration for corticostriatal connectivity	[138]
Optogenetic Stimulation of BDNF	Cortical Plasticity	↑ Activity-dependent BDNF, ↑ Decision-making accuracy (30%)	First optogenetic tool for precise BDNF modulation in brain networks	[139]
Flavonoid-Induced BDNF Enhancement	Cognitive Aging	↑ Plasma BDNF (25%), ↓ Cognitive decline (35%)	Established dietary polyphenols as natural BDNF modulators	[140]

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
