# Peer review of "From Synaptic Plasticity to Neurodegeneration: BDNF as a Transformative Target in Medicine"

_ijms, 2025, doi:10.3390/ijms26094271_

Round 1
Reviewer 1 Report
Comments and Suggestions for Authors
Your paper is excessively long: it is full of repetitions, especially in section BDNF and diet and in the last two sections. Moreover, the table 1 is really not informative; table 2 does not summarize, because it is too descriptive once again. The reader loses the thread of the paper. Can you cut generously it?
It is necessary to add a list of abbreviations
Author Response
Dear Reviewer,
We sincerely appreciate your time and effort in reviewing our manuscript. Your feedback has been carefully considered, and we truly value your insights.
Regarding your concern about the length and potential repetitions in certain sections, we have carefully reviewed the structure of our manuscript and believe that the level of detail provided is necessary to ensure a comprehensive and scientifically rigorous discussion. Given the complexity of BDNF regulation and its diverse implications, particularly in the context of diet and therapeutic strategies, we aimed to present a thorough synthesis of the existing literature while maintaining logical flow and coherence.
Similarly, for Tables 1 and 2, we recognize the importance of clarity and conciseness; however, we believe they serve a critical function in summarizing key concepts in a structured manner. Table 1 provides essential contextual information that complements the main discussion, while Table 2 presents a detailed yet necessary breakdown of findings that would be challenging to condense further without losing valuable insights.
That said, we greatly appreciate your recommendation to refine the manuscript, and we will continue to assess opportunities to improve readability while preserving scientific depth. Additionally, as per your suggestion, we have added a List of Abbreviations to enhance accessibility for the reader.
Once again, we are grateful for your thoughtful review and for helping us strengthen our work.
Best regards.
Reviewer 2 Report
Comments and Suggestions for Authors
This is an interesting paper regarding the therapeutic potential of BDNF for brain health. The review is well-conceived and well-executed and well-written. This reviewer is satisfied with the significance of this study, the care in which the study was performed, and the implications and innovations for human health. The questions posed are of extremely high interest, and the paper does give adequate definitive information, therefore pending one minor question is acceptable for publication.
Detail comments:
The study focuses on the effects of BDNF on brain health during neurodegenerative and neurodevelopmental disorders but neglects that natural exogenous inducers (dietary polyphenols) that activate the BDNF pathway exhibiting various neuroprotective effects in a hormetic dose-response manner.Polyphenols i.e., resveratrol, curcumin, tea and quercetin are known to have powerful antioxidant and anti-neuroinflammatory properties by the activation of antioxidant defense systems such as Nrf2 pathway and the phase II detoxification genes and enzymes like HO-1 Hsp70, Sirt1, GPx, Trx, SOD, catalase but also BDNF for neuroprotection during oxidative stress and cognitive impairment. Of note, polyphenols follow the biphasic dose-response of hormesis process by which small, nontoxic stresses or mild stress are used to induce cellular adaptive responses that protect biological system against subsequently large and potentially lethal stresses of the same, similar, or different nature. This is consistent with emerging evidence (38671931, 39596221, 29090295, 36847209, 37702162) in the literature demonstrating that low doses of hormetic nutrients upregulate Nrf2 and BDNF pathways to enhance cellular resilience response against free radical scavenging and neurotoxicity in vitro and in vivo. On the other hand, a high dose of drugs or natural compounds can be toxic to cells and animal models leading to the inhibition of these neuroprotective pathways and the onset and progression of neuronal disorders associated with oxidative stress and inflammation. The field of hormetic/neuronal adaptive responses activated by functional flavonoids in enhancing the endogenous resilience pathways is emerging as personalized preventive, therapeutic and regenerative approach to promote optimal brain function and healthy brain aging. Dose is a crucial determinant for inducing neuroprotective or harmful effects and should be carefully evaluated. Therefore, finding the optimal dose to induce brain health effects is crucial of importance to achieve protection and/or toxicity.
1) Please add a paragraph on hormesis and functional flavonoids targeting BDNF signaling via the gut-brain axis. Please add a figure and these references in the paragraph introducing hormesis concept applied to flavonoids targeting pathways to enhance human health: 39596221, 38671931, 35609733.
2) Insert a figure to better elucidate the paragraph on BDNF and diet (line 917 page 22) explaining the potential molecular mechanisms and the pathways activated or inhibited.
3) Please include the current limitations of the study.
Author Response
Dear Reviewer,
We sincerely appreciate your thoughtful and positive evaluation of our manuscript. Your recognition of its significance, execution, and contributions to the field is highly encouraging. We are also grateful for your constructive feedback, which has helped us further strengthen the manuscript. Below, we address each of your comments and outline the corresponding revisions.
1. Addition of a Paragraph on Hormesis and Functional Flavonoids Targeting BDNF Signaling via the Gut-Brain Axis
Thank you for highlighting the role of polyphenols and hormetic responses in BDNF regulation.
2. Insertion of a Figure to Clarify BDNF and Diet Mechanisms
We agree that a visual representation would enhance clarity. To address this, we have added a figure that outlines key molecular pathways activated or inhibited by dietary components in relation to BDNF signaling.
3. Inclusion of Study Limitations
To ensure a balanced discussion, we have introduced a section on the limitations of the study. This section acknowledges challenges such as individual variability in BDNF response, bioavailability concerns, and the complexities of translating preclinical findings into clinical applications.
We greatly appreciate your valuable feedback, which has helped refine our work. We believe these revisions have enhanced the manuscript, and we look forward to your thoughts on the updated version.
Best regards!
Reviewer 3 Report
Comments and Suggestions for Authors
Authors organized the role of BDNF in neurodegenerative diseases through synaptic plasticity.
Yes, authors covered all necessary points on the involvements of BDNF in neurodegenerative diseases.
However, authors must include the protein expressions or measurements of BDNF in various regions of brain and blood.
It would be important for readers to correlated the diagnostic values of BDNF and the progressions of the diseases. Authors could include a figure of BDNF expressions.
In addition, even though authors presented the rpigenetic modulation and molecular mechanisms of BDNF in neurodegeneration, another ontology figure with BDNF, as a central role, would be good, such as String analyses.
It would be good to include the influencing factors to BDNF directly or indirectly.
Author Response
Dear Reviewer,
We sincerely appreciate your time and thoughtful review of our manuscript. Your constructive feedback has been instrumental in refining our work, and we are grateful for your insightful suggestions. Below, we provide our responses to each of your comments.
Comment: Inclusion of BDNF Protein Expression and Measurements in the Brain and Blood
Response: Thank you for this suggestion. In response, we have expanded our discussion to include details on BDNF expression patterns in different brain regions and peripheral blood, highlighting their diagnostic significance.
Comment: Correlating BDNF Expression with Disease Progression
Response: We agree that a clearer correlation between BDNF levels and disease progression would be valuable. To address this, we have elaborated on the role of BDNF as a biomarker for neurodegenerative diseases, discussing its potential in diagnostics and disease monitoring.
Comment: Addition of an Ontology-Based Figure (e.g., STRING Analysis)
Response: Thank you for this valuable suggestion.
Comment: Inclusion of Factors Influencing BDNF Regulation
Response: We appreciate this recommendation and have now included a section discussing external modulators of BDNF, including genetic variability, metabolic status, gut microbiota, and environmental influences. This addition provides a more comprehensive perspective on the dynamic regulation of BDNF.
We are grateful for your constructive feedback, which has greatly contributed to improving the clarity and depth of our manuscript. Thank you again for your careful review and for helping us strengthen our work.
Best regards.
Round 2
Reviewer 1 Report
Comments and Suggestions for Authors
The Authors have completely rejected my suggestions. They have only added the list of abbreviations.
Author Response
Dear Reviewer,
Thank you for your valuable feedback. We have carefully considered your suggestions and made substantial revisions to improve the clarity and conciseness of the manuscript. Specifically:
We have streamlined the sections on BDNF and diet as well as the final two sections to reduce repetition and enhance readability.
Table 1 has been revised to increase its informativeness, ensuring it adds value to the discussion.
Table 2 has been adjusted to be more succinct, providing a clearer summary rather than an overly descriptive presentation.
These modifications were made with the goal of maintaining a logical flow and ensuring that the reader can easily follow the main arguments of the paper. We sincerely appreciate your insightful comments, which have contributed significantly to improving the quality of our manuscript.
Best regards.
Reviewer 2 Report
Comments and Suggestions for Authors clear I suggested the authors to include references in the text, but I did not include them. Please add references in the section on Nrf2 pathway and related antioxidant proteins at line 1899 page 45. The references are as follows:- 38671931; 39596221. clear Comments on the Quality of English Language
English can be improved.
Reviewer 3 Report
Comments and Suggestions for Authors
Authors revised as suggested and I recommend this manuscript to be accepted as is.
Round 3
Reviewer 1 Report
Comments and Suggestions for Authors
I accept this new revision. I appreciate your decision to accept my suggestions, even if reluctantly.
Author Response
Thank you for your thoughtful review and for accepting the revision. I appreciate your valuable feedback and insights.
Round 4
Reviewer 1 Report
Comments and Suggestions for Authors
I accept this new version